

# HydroBlocks-MSSUBv0.1: A Multiscale Approach for Simulating Lateral Subsurface Flow Dynamics in Land Surface Models

Daniel Guyumus[1], Laura Torres-Rojas[2], Luiz Bacelar[1], Chengcheng Xu[1], Nathaniel Chaney[1]

[1]Department of Civil and Environmental Engineering, Duke University, Durham, NC, USA

[2]Program Program in Atmospheric and Oceanic Sciences, Princeton University, Princeton, NJ, USA

*Correspondence to*: Daniel Guyumus (daniel.guyumus.preciado@duke.edu)

**Abstract.** Groundwater is critical in the hydrological cycle, impacting water supply, agriculture, and climate regulation. However, current Land Surface Models (LSMs) often struggle to accurately represent the multiple spatial scales of subsurface flow primarily due to the complexity of incorporating sufficient and yet efficiently surface heterogeneity, which significantly

influences subsurface dynamics. Accurately modeling this heterogeneity requires substantial computational resources, often making it challenging to achieve in practice. This study introduces a multiscale approach to address this limitation. The approach leverages the hierarchical clustering scheme of the HydroBlocks model to define hydrologically similar areas that the model uses to capture local, intermediate, and regional flow dynamics within regional units, which interact laterally based on hydraulic gradients and soil properties. The proposed method is compared against a benchmark simulation with 1.4 million

modeling units —34 times the number of tiles in the multiscale experiment. The results show consistency in spatial distribution and a Pearson coefficient of correlation above 0.85 for the temporal variability of hydrological variables such as latent and sensible heat flux, surface runoff, and effective saturation at the root zone, demonstrating its ability to represent subsurface flow patterns adequately. The scheme, however, struggles to adequately represent volumetric water content at the bottom of the soil column, as evidenced by lower correlation coefficients, where misrepresentation of elevation heterogeneity might play

a bigger role. This multiscale approach offers a computationally efficient way to incorporate detailed subsurface processes into large-scale hydrological simulations, improving our understanding of water cycle dynamics and supporting informed water resource management.

## 1 Introduction

Groundwater is critical in the hydrological cycle, impacting water supply, agriculture, ecosystems, climate regulation, drought

resilience, and overall sustainability (Gorelick & Zheng, 2015; Taylor et al., 2013). Its significance in Earth System Models (ESMs) relates to the dynamic interaction between the subsurface and the ocean, atmosphere, and the land surface. Subsurface flow influences surface water exchange, aquifers storage, and soil moisture redistribution, which significantly impact water and energy exchanges with the atmosphere and water balance in the vadose zone (Chen & Hu, 2004; de Graaf et al., 2015; Fan, 2015; Felfelani et al., 2021; Maxwell & Condon, 2016; Zeng et al., 2018).



Tóth's (1963) hierarchical classification categorizes subsurface flow into three primary flow systems: local, intermediate, and regional. Local flow refers to the shallow water movement within the same basin and hillslopes. In contrast, regional flow occurs at greater depths extending beyond superficial watershed boundaries and connecting large-scale recharge to discharge zones. Intermediate flow occurs along the continuum between the latter two flows.  This classification highlights how subsurface flow exhibits significant variations across spatial and temporal scales, influenced by landforms and subsurface

properties. These variations lead to diverse water table dynamics and surface water and energy fluxes (de Graaf et al., 2015; Fan, 2015; Freeze & Witherspoon, 1967). Studies indicate that lateral flow can substantially impact water table depths, especially in regions with heterogeneous soil properties or varying topography, as seen in North Africa, the Arabian Peninsula, and other areas experiencing water table decline (Maxwell et al., 2015; Rihani et al., 2010; Zeng et al., 2018). Lateral flow also affects the spatial distribution of soil moisture along hillslopes, influencing surface water and groundwater interactions

and, ultimately, maintaining the balance of groundwater systems. To accurately represent the hydrological cycle in Land Surface Models (LSMs), a comprehensive multiscale approach is essential to capture local to regional patterns of lateral subsurface flow.

The representation of subsurface dynamics in LSMs has advanced significantly in the last few decades due to the growth of high-performance computing and data availability. Multiple schemes to represent the two-way interaction between surface

water and the subsurface in LSMs have been developed and implemented (Bisht et al., 2018; Blyth et al., 2021; Fan et al., 2007; Fan & Miguez-Macho, 2011; Fisher & Koven, 2020; Hoch et al., 2023; Maxwell et al., 2015; Miguez-Macho et al., 2007; Miguez-Macho & Fan, 2012a, 2012b; Naz et al., 2023; Zampieri et al., 2012). These approaches vary from bucket and two-layered models to coupled groundwater frameworks and data-driven approaches. The first two types of approaches provide a simplified and computationally efficient representation of the subsurface system; the last two are more costly in data and

computation but can offer the highest level of detail by solving the complete three-dimensional equations to capture subsurface dynamics better (Guay et al., 2013; Jing et al., 2018; Maxwell et al., 2014; Sulis et al., 2010).

Within a single basin, topography plays a critical role in subsurface fluxes. Elevation differences between valleys and ridges create hydraulic gradients, the primary drivers of subsurface flow (Cardenas & Jiang, 2010; Harvey & Bencala, 1993; Tóth, 1963). However, accurately capturing these dynamics in LSMs presents a challenge. Current LSM grids typically operate at

coarse scales (tens to hundreds of kilometers), which are incompatible with the fine-scale variations in topography that govern local flow paths. Most LSMs also focus only on vertical flow processes within the soil column, and lateral flow is often neglected or simplified (Condon et al., 2021; Ji et al., 2017; Lam et al., 2011; Mendoza et al., 2015; Naz et al., 2023; Shrestha et al., 2015).

The modeling community has widely recognized the challenges of incorporating high-resolution to represent small-scale

subsurface processes within large-scale simulations (Choi, 2013; Fisher & Koven, 2020; Zhang & Montgomery, 1994; Zhao & Li, 2015). Several modern LSMs can numerically simulate subsurface flow dynamics with high spatial resolution. These models can explicitly represent variations in topography, soil properties, and vegetation cover, leading to more accurate simulations of water movement within a basin (Kollet & Maxwell, 2008; Maxwell et al., 2015). However, these high-resolution



simulations are often computationally expensive and impractical for large-scale applications like ESMs. The complexity of

solving three-dimensional flow equations across large extents with high spatial resolution pushes computational resources to

their limits.

One promising strategy for accounting for small-scale processes in large-scale simulations involves implementing tiling schemes. These schemes essentially group together hydrologically similar areas based on common characteristics such as soil type, land cover, topography, etc. (Flügel, 1995) that will respond similarly under equivalent meteorological forcing within an

LSM grid cell, reducing the overall number of modeling units and leading to improved computational efficiency (Chaney et al., 2016, 2018, 2021; Manrique-Suñén et al., 2013; Melton et al., 2017; Pilotto et al., 2017). This allows the incorporation of small-scale heterogeneity within large-scale simulations without sacrificing computational feasibility. Tiling schemes can bridge the gap between computationally expensive high-resolution models and practical large-scale simulations (Blyth et al., 2021; Clark et al., 2015; Fisher & Koven, 2020). Furthermore, a potential advantage of tiling schemes is that they simulate

lateral subsurface movement by allowing water exchange between neighboring tiles. This exchange can be based on factors like the hydraulic gradient or other relevant parameters specific to the hydrological setting. Moreover, tiling schemes offer scalability: the number and complexity of tiles within a grid cell can be adjusted based on the model's needs and available computational resources. This allows for efficient simulations at both coarse and fine scales.

This study addresses the need for tiling schemes to represent the lateral interaction of subsurface flow within LSMs at various

spatial scales. Specifically, it incorporates intermediate and regional flow into an existing modeling framework, HydroBlocks, leveraging its hierarchical clustering scheme to define tiles that interact laterally to account for local lateral flow. These model tiles are then grouped into regional units that can interact laterally across different spatial scales. The properties governing water movement within the new regional units are determined by the combined characteristics of the tiles (i.e., soil hydraulic properties), weighted by their relative area coverage. The proposed methodology is compared against a benchmark of the base

HydroBlocks model with a significantly larger number of tiles. This approach aims to represent a fully distributed solution that facilitates tile interaction between parallel processes and regional flows throughout the entire simulation domain. This comparison shows that the new multiscale scheme for lateral flow converges toward the synthetic truth after a long-term simulation. Furthermore, our multiscale scheme is relatively efficient in terms of computational requirements. A 50-year simulation for a 1x1° domain using high spatial resolution (~30 m) data with 40,700 tiles and a constant soil column depth of

10 m divided into six variable-depth layers takes 1,820 core-hours to run, producing hourly simulated time series. In comparison, the benchmark simulation with 1.4 million units takes approximately 8,147 core hours.

## 2 Data and Methods



## 2.1 Study area and input Datasets

Figure 1 shows the selected study domain located in Southwest Colorado in a 1°×1° box. The area, extending approximately 8,750 km², contains the headwaters of the Gunnison River drainage area, characterized by a heterogeneous hydrological system influenced by elevation, precipitation, and land cover. The river network includes the Taylor River, East River, Tomichi Creek, and the Blue Mesa Reservoir. The elevation within the domain ranges between 2,100 and 4,400 m above sea level (asl), with the main rivers draining west (Figure 1b). Land cover classification indicates mostly barren and sparsely vegetated areas in the

highlands, as shown in Figure 1c. Lastly, clay content at the surface suggests low permeability soils near the stream areas and higher permeability towards the ridges (Figure 1d), benefiting surface drainage and nutrient retention. Precipitation patterns vary significantly across the domain, with a declining gradient from west to east, approximating 496 millimeters per year over the entire region (Figure 1e).

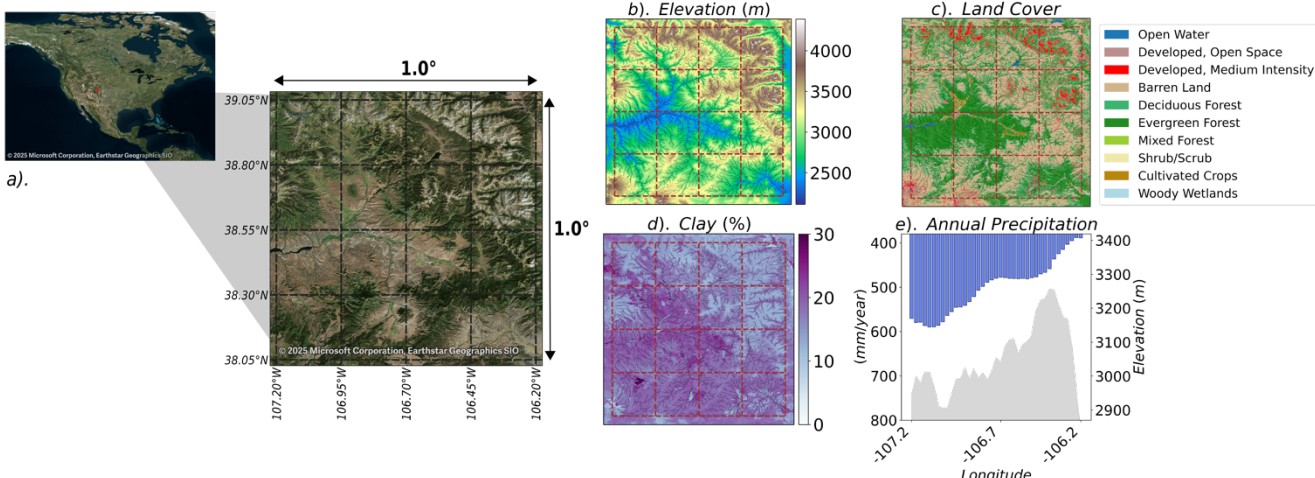

**Figure 1: Study domain in Colorado, USA, in a 1°×1° box. a) Satellite image of the study area divided by a 0.25° width grid showing the headwaters of the Gunnison River catchment (from Bing Maps, Microsoft. Retrieved 1 March 2025). b) Elevation from the National Elevation Dataset (NED) and stream network. c) Land cover classification from the National Land Cover Database (NLCD) with predominant barren areas. d) Soil clay content shows higher concentrations in the riparian zones and west-draining catchments. e) Mean annual precipitation shows higher rates towards the west combined with lower elevations tributary catchments.**

We used this study domain to test the representation of key hydrological processes at local, intermediate, and regional scales by allowing lateral subsurface water exchange between simulation units within the basin and between adjacent basins within the HydroBlocks framework. Furthermore, our proposed methodology can be scaled to conduct simulations at continental domains, following the same connectivity principles between parallel processes in High-Performance Computing (HPC) systems.

In this study, we used high spatial resolution and publicly available databases to parametrize Hydroblocks and explore the validity of the proposed multiscale methodology. The experiments were run over the study domain at an hourly time step over





50 years of simulation time, with the first 40 years used as the spin-up time. The data sets for topography, soil features, meteorology, and land cover used to drive HydroBlocks are described below.

*Elevation*: The National Elevation Dataset (NED; now known as 3DEP), maintained by the USGS, compiles comprehensive
elevation data for the United States and surrounding areas. Data sources for this dataset include various USGS topographic surveys supplemented by lidar collections, high-resolution aerial photography, and other public and private sources (Gesch et al., 2018). The NED30 (~30-meter resolution) digital elevation model was sink-filled and used to define the river network and drainage areas for the simulation domain.

*Soil Properties*: For the subsurface processes, we considered a 10 m deep soil column everywhere divided into six layers of
variable depths: 0.05 m, 0.1 m, 0.35 m, 0.5 m, 2 m, and 7 m. To parametrize the soil column, we used POLARIS (Chaney et al., 2019), a database of ~30-meter probabilistic soil property maps over the contiguous United States, covering variables such as soil texture, organic matter, pH, saturated hydraulic conductivity, and water retention curve parameters up to 2 m deep – for deeper soil layers, we used linear extrapolation.

*Meteorology*: Atmospheric forcing data, including precipitation, air temperature, radiation, and wind speed, over the
contiguous United States (CONUS) was obtained from the Princeton Climate Forcing (PCF) dataset at a 1/32° spatial resolution (~ 3 km) product at an hourly temporal scale from 2010 to 2019 (Sheffield et al., 2006).

*Land Cover*: Land cover data was obtained from the National Land Cover Database (NLCD) at ~30-meter resolution, a product based on a modified Anderson Level II classification system and 16 classes (Homer et al., 2015).

## 2.1 Land Surface Model: HydroBlocks v0.2

HydroBlocks v0.2 (Chaney et al., 2021) is a land surface modeling framework that addresses the challenges associated with accurately representing sub-grid heterogeneity in LSMs. The model effectively clusters spatial covariates for soil properties, topographic features, and meteorological variables to reduce the computational expense of resolving a high spatial-resolution LSM (Chaney et al., 2016). This is important, as a failure to represent sub-grid heterogeneity can lead to potential errors in simulated water fluxes and ecosystem dynamics, especially in areas with steep terrain or high spatial variability in soil
properties (Clark et al., 2015; Fisher & Koven, 2020; Giorgi & Avissar, 1997; Torres-Rojas et al., 2022).

The Hierarchical Multivariate Clustering scheme (HMC) employed by HydroBlocks to derive tiles from available high-spatial resolution datasets is a key feature that significantly reduces the number of computational units while preserving essential local scale processes within large-scale simulations. Figure 2 shows the main steps of HydroBlocks' HMC algorithm. First, the domain is divided into macroscale polygons for parallel computing. Each subdivision comprises self-contained watersheds
that are then grouped into *k* clusters of watersheds based on specific watershed-aggregated spatial covariates. Each cluster of watersheds is further discretized into height bands to differentiate between streams, valleys, and hilltops; for this step, every subsequent high band area is *n* times larger than the previous one, providing higher discretization near the streams. Finally, an average of *p* intra-band clusters is created for each high band based on soil parameters and land cover covariates representing field-scale heterogeneity (Chaney et al., 2021). It is important to acknowledge that the tiling structure, as pointed out in Torres-




Rojas et al., 2022b and Chaney et al., 2016b, obtained through the clustering mechanism –parameters *k, p,* and *n*– as well as the chosen proxies of spatial heterogeneity will influence how close the model approximates the complexity and heterogeneity of the hydrological processes for the specific domain.

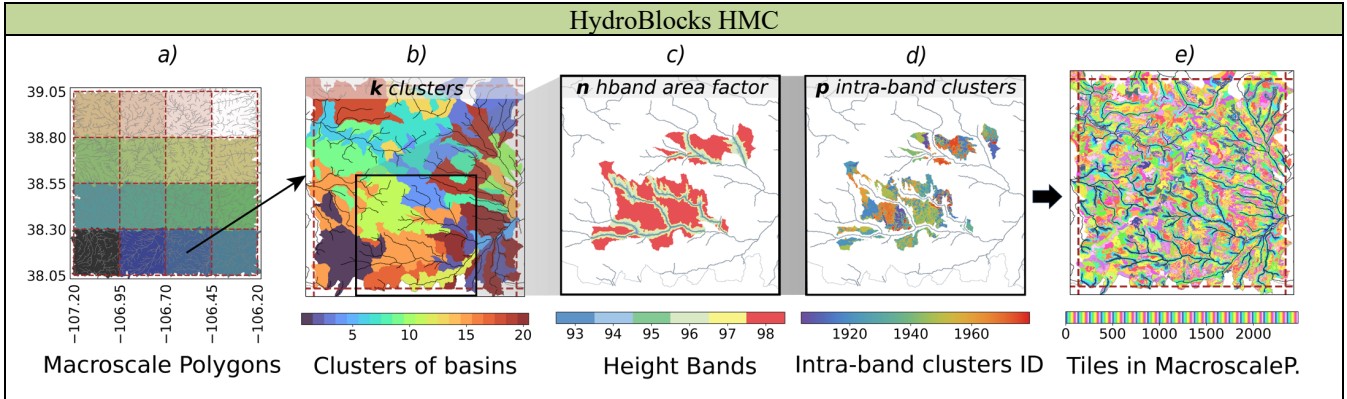

**Figure 2: HydroBlocks HMC algorithm. a) the domain is divided into 16 self-contained macroscale polygons; each color indicates a unique subdomain for parallel computing. b) macroscale polygon #3 is discretized into *k=20* watershed clusters using large-scale**
**heterogeneity proxies (latitude, longitude, drainage area). c) cluster of watersheds #18 in the highlighted area in frame b) is divided into height bands where the area of each subsequent band is *n=2* times larger than the preceding one. d) each band is further discretized on average *p=20* intra-band clusters across the height bands using proxies of field scale spatial heterogeneity (land cover, elevation, latitude, longitude, clay content); this frame shows the computed tiles for the uppermost height band. e) the discretization is applied to each cluster of watersheds; consequently, each cluster contains a unique set of tiles. For this case, a total of 2,480 tiles**
**are defined for macroscale polygon #3.**

### 2.3 Noah-MP

HydroBlocks uses the vertical 1D-column model Noah-MP (Niu et al., 2011), a comprehensive land surface resolving framework used to characterize hydrological and energy processes over land and the subsurface using a tile structure. The model includes physical representations of dynamic vegetation, canopy interception, multi-layer snowpack physics, and soil

and hydrological processes. A simple groundwater model based on a TOPMODEL runoff scheme is also implemented to account for critical zone influence and long-term memory of antecedent weather events (Niu et al., 2005, 2011). When combined with HydroBlocks, Noah-MP simulates various groundwater processes, including recharge, water table change, and baseflow.

At each time step, the model simulates soil moisture and vertical water flow in unsaturated soils across the soil layers using
the one-dimensional Richards equation (Eq. 1) to update the hydrological states of each tile. Within HydroBlocks, the layers of the tiles interact laterally via dynamic subsurface flow between adjacent neighboring areas, computed using Darcy's flux (Chaney et al., 2021).

$$\frac{\partial \theta}{\partial t} = \frac{\partial}{\partial z}\left(D(\theta)\frac{\partial \theta}{\partial z}\right) + \frac{\partial K(\theta)}{\partial z} - S \,, \tag{1}$$

Where $\theta$ is the volumetric soil moisture content, $D$ is the soil water diffusivity, $K$ is the hydraulic conductivity, $S$ represents
sources and sinks of soil moisture, $t$ is the time variable, and $z$ indicates the vertical coordinate.





(2)

In Eq.1, $S$ accounts for soil surface infiltration, water loss due to plant transpiration, and surface evaporation. For the groundwater model, a TOPMODEL-based runoff scheme with an equilibrium water table was chosen (Niu et al., 2005), setting a no-drain condition at the bottom layer. However, water can be removed as lateral flux between soil columns as a new

source/sink term $G$ introduced in the HydroBlocks framework (Eq. 2).

$$\frac{\partial \theta_i}{\partial t} = -\frac{\partial q_z}{\partial z_i} - S_i - G_i, \qquad (2)$$

Here, the new source/sink term $G_i$ describes the computed fluxes —assuming Darcy's law— between adjacent tiles as the divergence at the $i$-th layer, as shown in Eq. 3.

$$G_i = -T_i \frac{\Delta h_i}{\Delta x} \frac{w}{A_T} \frac{1}{\Delta z_i}, \qquad (3)$$

Where $T_i$ is the harmonic mean transmissivity between the interacting layers of two different tiles, $\Delta h_i$ is the hydraulic head gradient at the $i$-th layer, $\Delta x$ is the horizontal distance between the centroids of the tile's polygons, $w$ is the length of the shared perimeter between the tiles, $A_T$ is the surface area of the corresponding polygon, and $\Delta z$ is the thickness of the soil layer.

## 2.4 Lateral subsurface flow: Multiscale Scheme (HydroBlocks-MSSUBv0.1)

The divergence term described in Section 2.3 accounts for lateral flow at a local scale, given that the tiles interact laterally

within the same cluster of watersheds. However, a term that accounts for intermediate and regional flow paths has yet to be included. Here, we have developed a scheme to compute lateral flow for these longer flow paths following the same mechanism currently implemented in HydroBlocks to compute local lateral flow. To achieve this, we used the clusters of watersheds as Regional and Intermediate Subsurface Flow Units (RISFU) and a soil column underneath that interacts laterally with the neighboring areas at every layer, following Darcy's law to update the hydraulic state variables of the tiles within.

Figure 3 illustrates the proposed multiscale scheme for simulating lateral subsurface flow using HydroBlocks. The methodology involves five main stages: (1) Domain discretization into macroscale polygons and corresponding local tiles (using HydroBlocks' HMC), (2) Defining Intermediate and Regional Units (RISFU) to represent the subsurface system, (3) computing aggregated Soil Hydraulic Properties to parametrize the RISFUs, (4) Computing subsurface lateral flow at different levels (regional, intermediate, and local), and (5) Updating vertical flow of the soil column within Noah-MP. Stages 1 and 2

are performed before the start of the simulation time, and Stages 3 to 5 are repeated for each timestep until the final simulation time step is reached. Details on each stage are presented below.





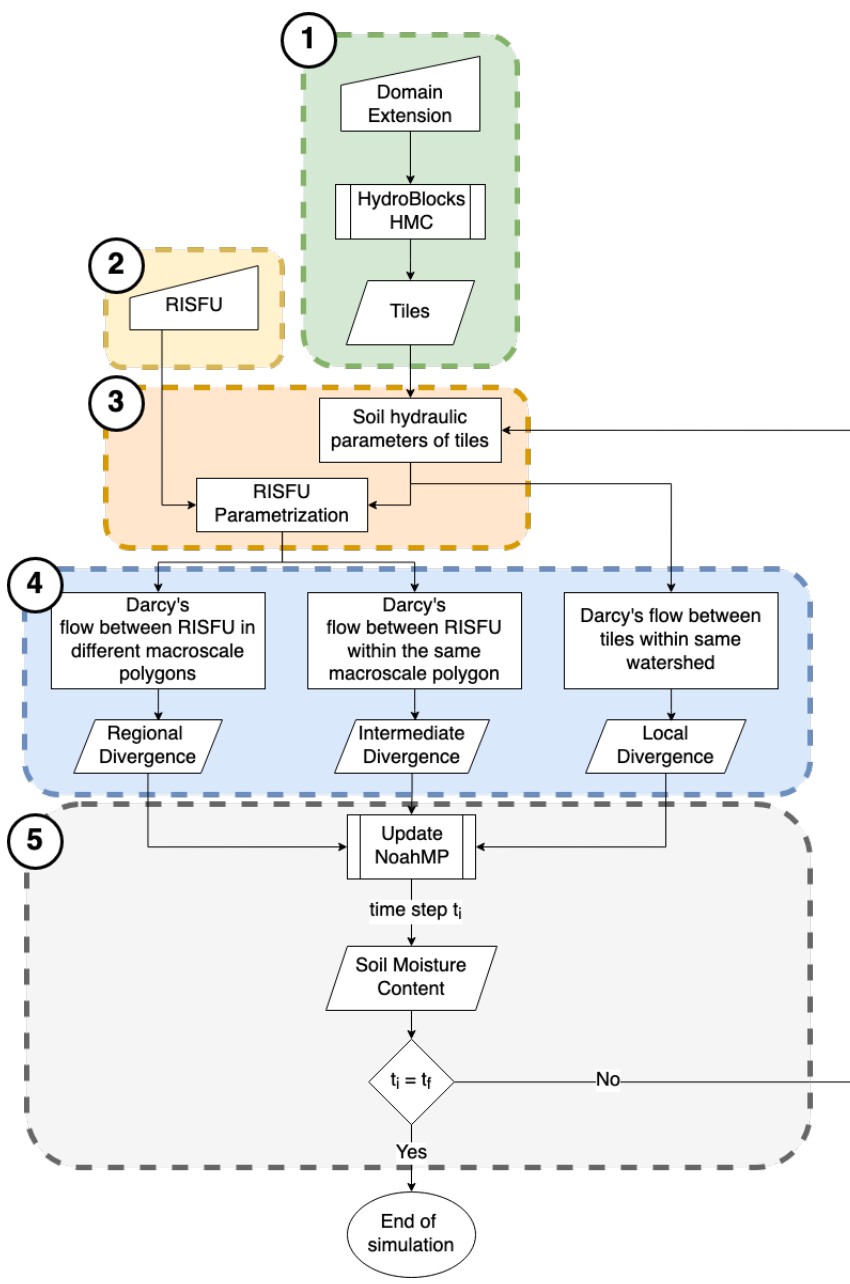

**Figure 3: Figure 3. Proposed Multiscale Methodology flow chart. Step 1 consists of HydroBlocks' HMC to generate tiles per macroscale polygon. Step 2 is the generation of the RISFUs from the cluster of watersheds obtained through the HMC algorithm. Step 3 is the parametrization of the soil hydraulic properties of the RISFUs using the tiles inside them. Step 4 consists of computing the subsurface flow between the RISFUs within the same macroscale polygon (*intermediate flow*) and between RISFUs along the borders of different macroscale polygons (*regional flow*); *local flow* is the flow between tiles within the same cluster of watersheds as is already implemented in HydroBlocks. Step 5 updates the divergence term in Richard's equation resolved with Noah-MP to include the three scales of subsurface flow for the next time step $t_i$ until the end of the simulation $t_f$ is reached.**



**Step 1. Domain discretization**

As mentioned in Section 2.2, HydroBlocks' HMC initially divides the modeling domain into macroscale polygons to allow for efficient parallelization in High-Performance Computing systems. Each polygon represents a self-contained group of basins within a predefined bounding box. Subsequently, the HydroBlocks HMC algorithm generates a set of independent tiles, as described in Section 2.2 (see Figure 2). These tiles serve as simulation units in the Noah-MP model, enabling the solution of water and energy fluxes.

**Step 2. Definition of regional and intermediate subsurface flow units (RISFU)**

Once each macroscale polygon has its own set of tiles, the next step involves discretizing the macroscale polygon on RISFUs (Figure 4). For this experiment, these intermediate and regional units are chosen to be the same clusters of watersheds obtained from the intermediate step in the HMC algorithm. Consequently, each macroscale polygon is subdivided into a finite number of RISFUs representing the intermediate and regional-scale subsurface system's structure. The RISFUs do not need to be the same clusters of watersheds, and the extension, shape, and number can be modified to meet specific criteria, making this approach flexible for its implementation in large-scale simulations. For our study case, the number of clusters of watersheds generated and, consequently, the number of RISFUs is determined by the parameter $k$ chosen equal to 20. It is worth mentioning that the selection of parameter $k$ and the proxies of large-scale heterogeneity will prescribe how the subsurface hydrological structure is represented in the LSM.

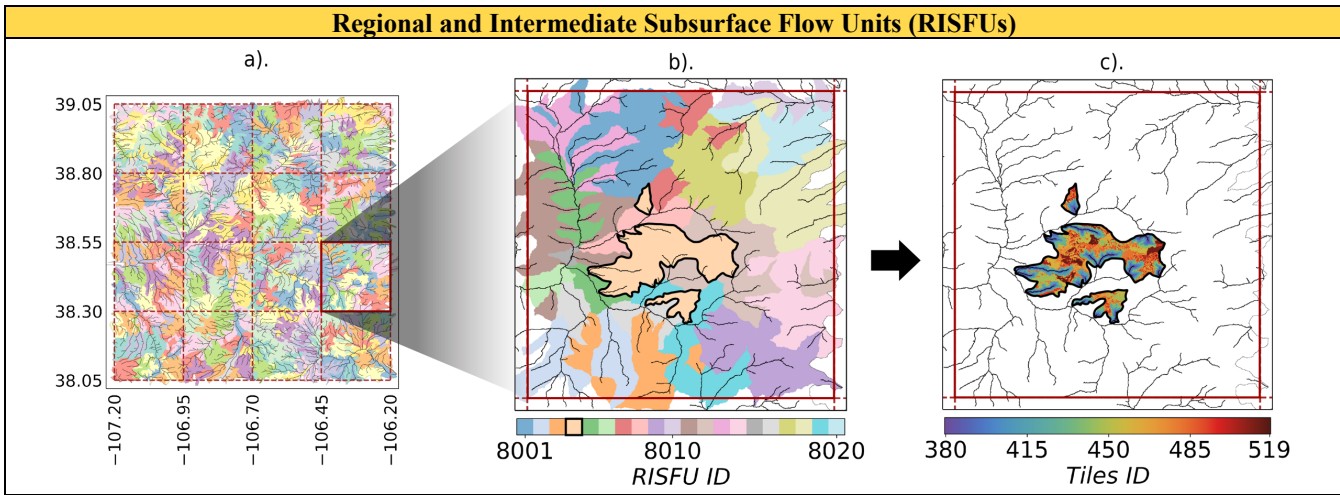

**Figure 4: Regional and Intermediate Subsurface Flow Units (RISFUs). a) Map of clusters of watersheds obtained from the HMC algorithm. For this experiment, the parameter *k=20*. b) Each macroscale polygon contains a total of 20 RISFU, chosen to be the same as the clusters of watersheds in frame a). The outlined area is a single RISFU. c) the HydroBlocks tiles within the isolated RISFU from frame b) that are used to compute the matrix of weights in Step 3a.**





**Step 3. Soil parameters Aggregation**

The soil hydraulic properties of every layer of the RISFUs are determined by calculating the weighted average of the soil hydraulic properties from the corresponding layers of the individual tiles within each unit (as shown in Figure 5 for saturated hydraulic conductivity ($K_{sat}$). The weights are set to be the fraction of the RISFU's area covered by each tile. This weight-averaging approach is applied to soil hydraulic parameters and state variables, including soil moisture, saturated water content, residual water content, and saturated hydraulic conductivity. By effectively combining the hydraulic properties of the constituent tiles, we create a representative characterization of the soil column for each RISFU. The procedure used to perform the aggregation of properties is detailed below.

a) First, we compute the fraction of the area that each tile covers of the RISFU $i$. Each component $a_{ij}$ represents the fraction of RISFU area $A_R$ covered by the area $A_T$ of tile $j$ within that specific unit $i$ to generate the matrix of area fractions $[A]$ (Eq. 4).

$$[A] = \begin{bmatrix} a_{11} & a_{12} & \cdots & a_{1j} \\ a_{21} & a_{22} & \cdots & a_{2j} \\ \vdots & \vdots & \ddots & \vdots \\ a_{i1} & a_{i2} & \cdots & a_{ij} \end{bmatrix}, a_{ij} = \frac{A_{T_j}}{A_{R_i}}, \tag{4}$$

b) The RISFU's soil hydraulic properties (and state variables) $P_R$ are computed as the weighted average of those of the tiles within the RISFU boundaries $P_T$. The process is repeated for each soil layer (Eq. 5).

$$P_R = [A]P_T, \tag{5}$$

c) Every time step, the Soil Hydraulic Conductivity ($K$) of each RISFU layer is computed following the Brooks-Corey retention curve (Eq. 6).

$$K(\theta) = K_{sat}\left(\frac{\psi}{\psi_b}\right)^{-2-3\lambda}, \tag{6}$$

Where $K_{sat}$ is the Saturated Hydraulic Conductivity, $\psi$ is the matric potential, $\psi_b$, $\lambda$ are model parameters, and $\theta$ is volumetric water content defined in Eq. 7 as:

$$\theta = (\theta_s - \theta_r)\left(\frac{\psi_b}{\psi}\right)^{\lambda} + \theta_r \tag{7}$$

Where $\theta_s$ and $\theta_r$ are the saturated and residual water content respectively.

| Tile Aggregation of Soil Parameters |
|:---:|





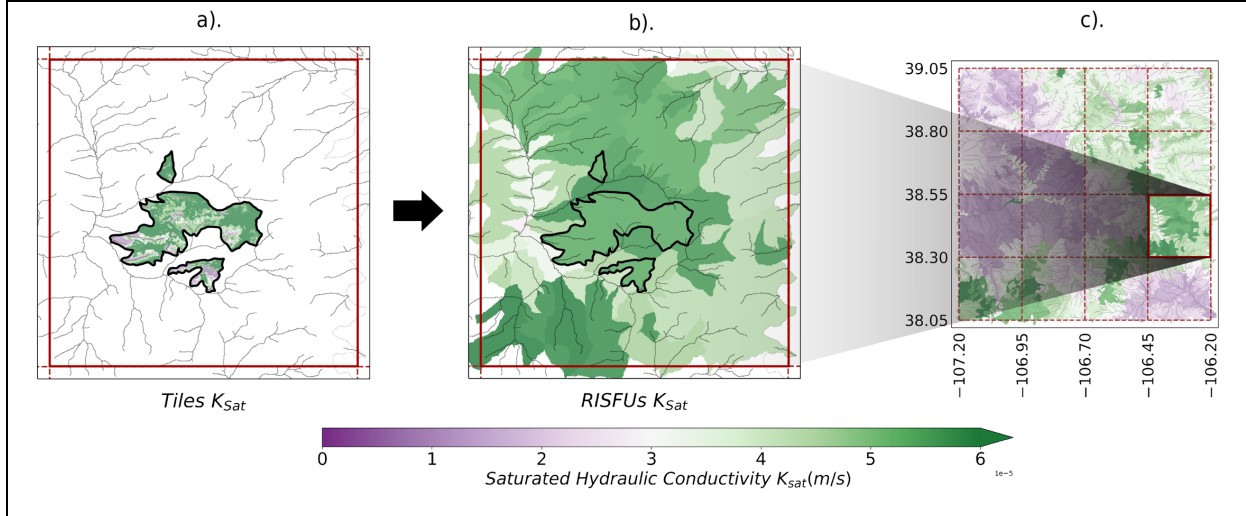

**Figure 5: Tile aggregation of soil hydraulic parameters to characterize RISFUs. a) Saturated Hydraulic Conductivity ($K_{sat}$) map of tiles within a single RISFU. Values are for the root zone layer. b) Weighted-average $K_{sat}$ for each RISFU (root zone layer). The process is repeated for all 20 RISFU in the macroscale polygon and for each layer of the soil column. c) The process is then repeated across all 16 macroscale polygons to generate the map of the 1×1° domain with the characterized RISFUs.**

**Step 4. Subsurface Lateral Flow**

For *local subsurface flow* (i.e., inter-tile level), HydroBlocks computes lateral flow between tiles connected within the same watershed using Darcy's law as introduced in Eq. 3.

To calculate the distance in the direction of the flow, each tile is approximated to a rectangular shape of area $A_T = b\,w,$ as shown in Figure 6, where $b$ is the side of the rectangle to be solved and $w$ is the length of the shared perimeter between tiles, equal to the sum of the width of all grid cells in contact with the neighboring tile. This assumes that the flow in/out of the tile is homogeneous along the shared length and gets distributed across the entire area instantaneously, allowing us to define the horizontal traveling distance $\Delta x$ for one of the tiles as half of the side $b$ (Eq. 8).

$$\Delta x = \frac{b}{2}; b = \frac{A_T}{w}, \tag{8}$$

We used the same approach described above to compute the horizontal distances between centroids of RISFUs. These areas are also approximated to a rectangular shape, and horizontal distances and shared lengths are computed; this is done for the RISFU within the same macroscale polygon and for the RISFU at the perimeter in contact with adjacent macroscale polygons.





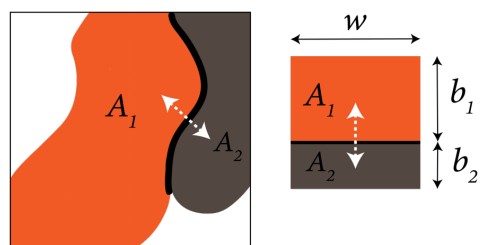

**Figure 6: Area approximation for irregular tiles and RISFUs. Both tiles and RISFU are approximated to a rectangle of area $A = b\,w$. Where $w$ is the shared perimeter length between two tiles or RISFUs. The horizontal traveling distance in the direction of the flow is then computed as $\Delta x = b_1/2 + b_2/2$**

The flux between tiles within the same basins (*local scale*) and between RISFU units (*regional/intermediate scale*) is computed for each time step, assuming Darcy's flow, as presented in Eq. 9. The net flow specifies the divergent subsurface flow for each tile or RISFU driven by the surrounding areas of the same scale.

$$Q_R = \mathrm{diag}\left(-[K_R]\frac{[\Delta h]}{[\Delta x]}\right), \qquad\qquad (9)$$

Where $Q_R$ is the net flow per RISFU, $K_R$ is the harmonic mean of hydraulic conductivity between connected units, $h$ and $x$ are the hydraulic gradient and horizontal distance respectively.

Figure 7 illustrates the process of computing lateral flow at both intermediate and regional scales. Figure 7a presents a schematic example of four macroscale polygons with six interacting RISFUs. At a specific time step, the hydraulic gradient determines the subsurface flow (q) in the direction of the red arrows. The subscript denotes the interacting units, and the sign indicates the direction of the flow. For q2,1 and −q2,1, the magnitude of the flow is equivalent, with the positive sign indicating flow from RISFU 2 to RISFU 1 in the former case. The net flow Q is then computed as the sum of all the flows (q) per unit. This is repeated for each layer of the soil column.

Note that the boundaries of the macroscale polygons are purely schematic to illustrate an approximate division. However, the actual perimeter of the macroscale polygons follows that of self-contained watersheds, ensuring that no RISFU is divided between macroscale polygons.

**Lateral Flow between RISFU within and across macroscale polygons**





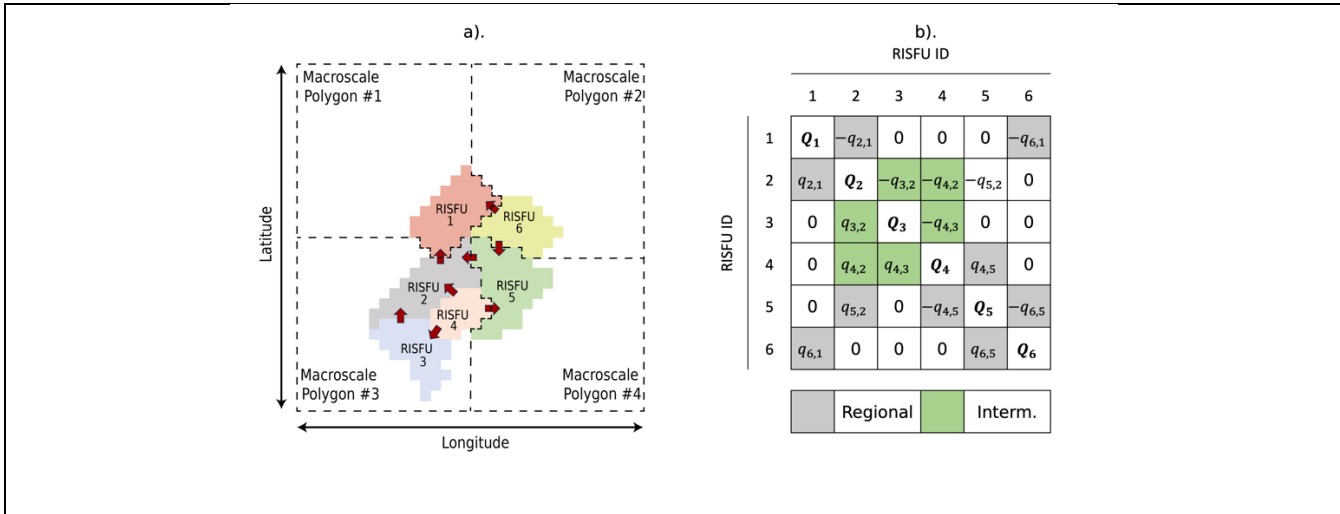

**Figure 7: Lateral flow scheme for intermediate and regional scales. a) An example of six RISFUs at four different macroscale polygons to visualize the process of computing net lateral flow between RISFU. The direction of the flow at a specific time step is indicated by the red arrows. The perimeter of each macroscale polygon is represented by the dashed line. b) Matrix of flows between interacting units. In grey are the flows between units at different macroscale polygons to represent regional scales, and in green are the intermediate flows. Units that are not laterally connected are assigned a zero flow. The net flow for each RISFU is then computed as the arithmetic sum of all the flows $q$ in the same row.**

The regional and intermediate net flows are redistributed among the tiles within each RISFU based on the matrix of areal fractions to obtain a single value for each tile, as shown in Eq. 10.

$$Q_T = [A]Q_R, \tag{10}$$

**Step 5. Update Land Surface Model**

The net flow per each tile $T$ is then converted to a divergence $G$ to be used as a source/sink term in the vertical solution of the differential equation in NOAH-MP following Eq. 11. The process is repeated each time step from stages 3 to 5 (see Figure 3) using the solution for soil moisture content to update the hydraulic conductivity and hydraulic gradient until the simulation is completed.

$$G_T = Q_T \frac{1}{A_T \Delta z}, \tag{11}$$

Where $A_T$ is the surface area of each tile, and $z$ is the layer's depth.

Following Eq. 2, where a new divergence term is added to include local lateral subsurface flow, the presented multiscale scheme updates the vertical solution of soil moisture with two extra divergence terms to represent intermediate and regional scales, shown in Eq. 12. The superscripts $(R)$ $(I)$ $(L)$ indicate the Regional, Intermediate, and Local levels of each divergence term.

$$\frac{\partial \theta}{\partial t} = -\frac{\partial q_z}{\partial z} - G_T^{(R)} - G_T^{(I)} - G_T^{(L)}, \tag{12}$$



## 2.5 Experimental Setup in HydroBlocks

To test our approach, we performed a set of experiments aimed at developing and validating an enhanced multiscale

HydroBlocks framework designed to better capture water movement across different scales—ranging from local to regional. Our approach includes the extra two divergence terms described in Section 2.4 implemented within the tiling framework. We aim to evaluate the logical consistency of our scheme by selecting a 1°×1° domain, and applying HydroBlocks' HMC (see section 2.2). The experiments were run at hourly time steps for a total of 50 years.

As illustrated in Figure 8, the study area was partitioned into 16 macroscale polygons to facilitate parallelization. The domain

is discretized into 40,700 tiles derived from high-resolution environmental data to capture the small-scale heterogeneity in elevation, land cover, and soil properties. The clusters of watersheds obtained through HydroBlocks' HMC for each macroscale polygon are then adopted as RISFU to represent the intermediate and regional units to model lateral subsurface flow.

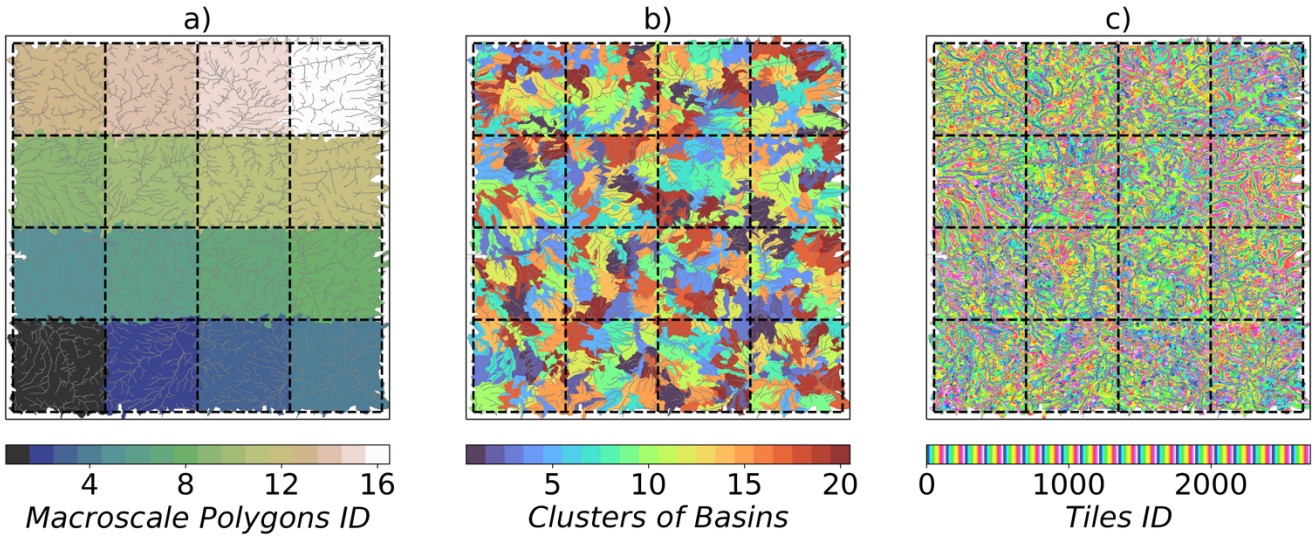

**Figure 8: Experimental setup. a) shows the domain divided into 16 self-contained macroscale polygons for parallel computing. b)**
**map of the cluster of watersheds considering *k=20* for the HMC algorithm in HydroBlocks. c) map of the tiles derived through the HMC algorithm in HydroBlocks; each macroscale polygon contains an average of 2,500 tiles. HydroBlocks' HMC generates a total of 40,700 tiles using parameters *k=20, n=2, p=20*.**

### 2.5.1 Test Case under Homogeneous Conditions

Initially, controlled conditions were used to isolate the influence of elevation on subsurface flow, testing our hypothesis that

the multiscale scheme will primarily show divergence flow patterns aligned with the topography.

For this experiment, variables, including land cover, soil properties, and meteorology, were homogenized across the 1°x1° domain to isolate the influence of elevation as the primary driver of subsurface flow. However, vertical heterogeneity remains between the layers of the soil column. This means that while each soil column within the domain's constituent tiles has uniform soil properties per layer, the properties differ between layers. This homogenization enables a focused evaluation of the new



multiscale scheme's capability to represent water movement at various scales. Figure 9 shows the RISFUs used in this experiment and the saturated hydraulic conductivity map as an example of one of the homogenized parameters used.

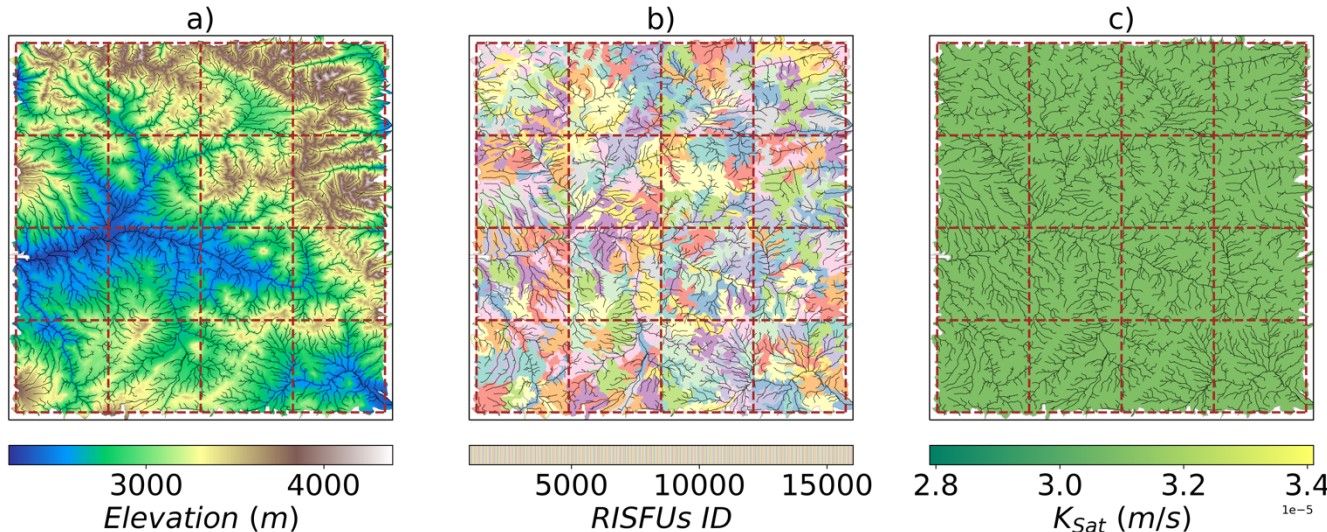

**Figure 9: Regional and Intermediate Subsurface Flow Units (RISFUs). a) shows the elevation at ~30 m scale resolution. b) the RISFUs generated based on the cluster of watersheds (Figure 8b), notice that each macroscale polygon has a total of 20 RISFUs. c)**
**Saturated Hydraulic Conductivity of the first layer of the soil column is presented as an example of the homogenous soil hydraulic properties for all 40,700 tiles. Land cover and meteorological forcing data were also homogenized.**

### 2.5.2 Test Case under Heterogeneous Conditions and Comparison against Baseline

The second part of this experiment introduces —layer by layer— the influence of small-scale heterogeneity from the land cover, soil hydraulic properties, and meteorological forcing variables to the homogeneous baseline presented in the previous
section; this is, we returned each tile to its original parameters set to account for a more realistic effect on water absorption, retention, runoff, and infiltration.

Lastly, a comparison between our multiscale scheme and the baseline HydroBlocks identified differences in soil moisture, runoff, latent heat, and sensible heat.

### 2.5.3 Comparison of the Multiscale Scheme against a HydroBlocks Benchmark

We compared the outputs of the long-term simulation from the previous experiment (heterogeneous conditions) against a HydroBlocks benchmark. This benchmark consisted of 1.4 million tiles distributed among 64 parallel processes, as shown in Figure 10. This configuration increased the number of tiles per macroscale polygon and the number of macroscale polygons, thereby capturing essential spatial heterogeneity and lateral subsurface interactions while remaining computationally feasible, given our constraints. The benchmark simulation also incorporated lateral interactions between the tiles located at the
perimeters of each macroscale polygon, enabling water movement across the entire domain to capture regional and intermediate flow patterns.




Ideally, we would compare our simulations against a fully distributed model, where every grid cell of approximately 30-meter resolution functions as its own tile (resulting in approximately 13 million tiles for the study area). However, due to computational constraints, simulating a 50-year-long hourly time series with such a detailed model was unfeasible with our

available resources. Therefore, we opted for the highest number of tiles feasible within our limitations. The selected number of 1.4 million tiles represents 11% of the number of tiles required for a fully distributed simulation.

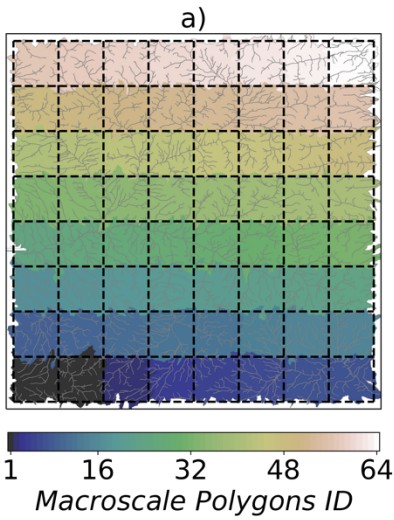
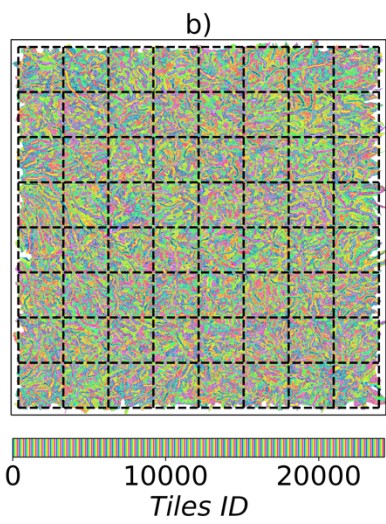

**Figure 10: Benchmark setup. a) the domain was divided into 64 macroscale polygons that were used to parallelize the LSM. b) HydroBlocks HMC algorithm created approximately 1.4 million tiles to represent the spatial heterogeneity across the domain; each**
**tile interacts laterally at the subsurface level with the neighboring areas within each macroscale polygon and with the tiles in the adjacent macroscale polygons if located in the perimeters.**

## 3 Results

### 3.1 Test Case under Homogeneous Conditions

This section delineates the results of our experiments used to evaluate the new multiscale framework. Our primary objective
is to assess the performance of the enhanced HydroBlocks framework, which incorporates *regional, intermediate*, additionally to the already implemented *local* lateral subsurface flow, and compare it to its baseline version, which only includes *local*. The homogeneous experiment was run at an hourly time step for ten years. Under these controlled conditions, the multiscale framework was able to reproduce water movement between the RISFUs across and within macroscale polygons and among tiles of the same watershed following the topography, which was expected to be the first-order driver of subsurface flow.

Figure 11 summarizes the results for the one-year hourly average of the divergence flow computed for three different scales: *regional, intermediate*, and *local*; negative divergence values (in blue) indicate subsurface inflow and positive divergence values (in red) indicate outflow from the RISFUs and/or from the tiles.





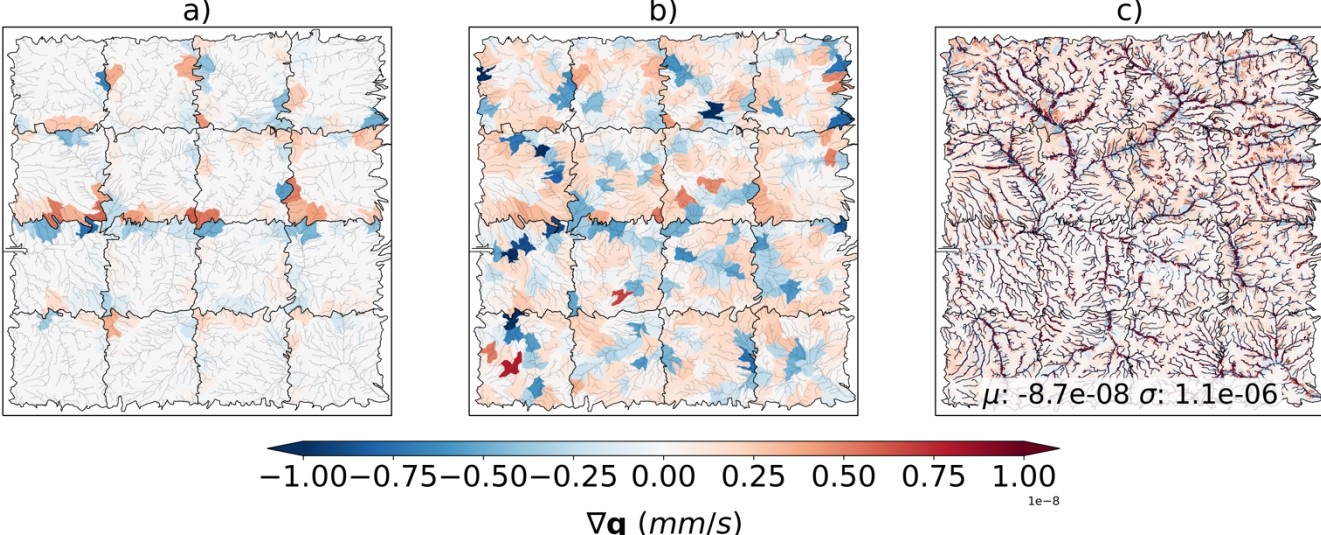

**Figure 11: One-year average divergent subsurface flow under homogeneous soil hydraulic properties and meteorological forcing data, the blue color gradient indicates negative divergence (water gain), and the red color gradient indicates positive divergence (water loss). a) Regional divergence subsurface flow among RISFUs at the boundaries of the macroscale polygons. b) Intermediate divergence subsurface flow among RISFUs within macroscale polygons in addition to the regional divergence from frame a). c) Local divergence subsurface flow among tiles within the same cluster of watersheds.**

*Regional subsurface flow:* Figure 11a illustrates the temporal mean divergence between RISFUs along the perimeters of macroscale polygons. Positive divergences are observed in higher elevation regions, indicating a net loss of water, whereas negative divergences in lower elevations suggest subsurface water gains across subdomains. This pattern aligns with the elevation map in Figure 9a, as RISFUs account for the average elevation of the constituent tiles that, under homogeneous soil properties and meteorological forcing, lead to hydraulic heads of the soil column to resemble the topography.

*Intermediate subsurface flow:* Figure 11b maps the average divergence among RISFUs within the macroscale polygons, plus the influence of regional subsurface flow at the boundaries. The subsurface flow distribution occurs among units within each 0.25° x 0.25° macroscale polygon. Consistent with Figure 11a, elevation remains the primary driver of water movement, with higher elevations draining subsurface flow towards lower elevations, leading to flow accumulation near riparian zones.

*Local subsurface flow:* Figure 11c integrates the regional (Figure 11a), intermediate (Figure 11b), and local subsurface flows — the latter derived from the HydroBlocks implementation of lateral interaction among tiles within the same watershed —. The regional and intermediate subsurface flows from the RISFUs are redistributed to the tiles, as detailed in Section 2.4, and the local divergence flow between tiles within the same watershed is added. The resultant divergence map clearly shows stream area tiles accumulating water while adjacent riparian zones exhibit water loss. Stream areas and riparian zones proximal to the main river also show greater water accumulation compared to ridge areas within the contributing watersheds.



Based on this initial analysis, we conclude that the implemented multiscale scheme effectively models regional and intermediate water movement across the domain by considering only elevation heterogeneity, while still remaining compatible with process parallelization. This outcome highlights the multiscale scheme's effectiveness in capturing the spatial dynamics of water redistribution for large-scale subsurface flow within the land surface model.

**3.2 Test Case under Heterogeneous Conditions and Comparison against Baseline of HydroBlocks**

Figure 12 shows the temporal average divergence of the subsurface flow (aggregated *regional, intermediate,* and *local* divergence) for the same domain in the three heterogeneous cases.

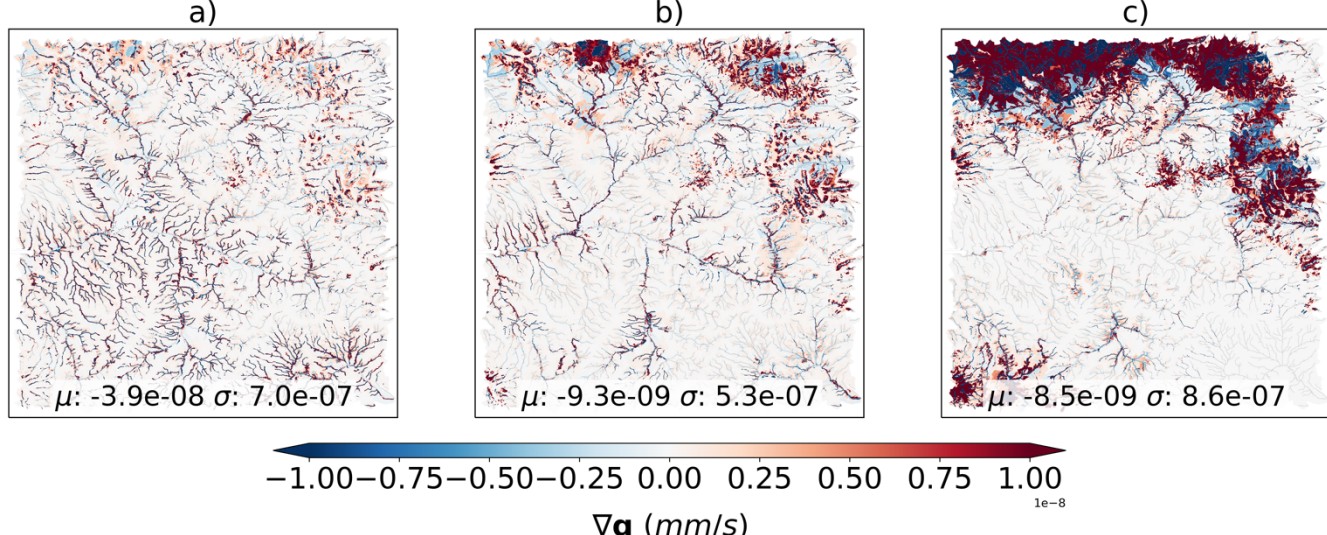

**Figure 12: Divergent subsurface flow under heterogeneous conditions. a) Time average divergent subsurface flow under heterogeneous land cover. b) Divergence map of the experiment under heterogeneous land cover and soil hydraulic properties. c) Divergence map under heterogeneous land cover, soil hydraulic properties, and meteorological forcing variables (precipitation, wind, etc.)**

*Heterogeneous Land Cover:* The first layer of heterogeneity is introduced by the spatial variability of land cover. Figure 12a illustrates the computed divergence flow, where a change of ~55% in the mean is observed compared to the homogeneous case in Figure 10c. This positive shift can be attributed to reduced runoff generation rates and enhanced water absorption in riparian zones due to vegetation.

*Heterogeneous Soil Properties:* The second layer of heterogeneity consists of the spatially variable set of soil properties for each tile's soil column. Figure 12b shows a positive change in the mean of ~120%, primarily near the riparian zones, indicating a slower contribution to subsurface flow rates. This is potentially due to the effect of soil texture, where soils with high water retention capacity slow down lateral water movement.

*Heterogeneous Meteorology:* The final layer of heterogeneity is meteorology. Figure 12c displays a more pronounced pattern shift in subsurface flows compared to the homogeneous case, with higher rates of positive divergence near the north of





the domain. In this instance, however, the change in the spatial mean is small (~9%), while the pattern difference in subsurface flow distribution is explained throughout the change in standard deviation (~60%) towards a more concentrated distribution

around the mean compared to the heterogeneous soil properties case. Higher divergence values are attributed to higher precipitation rates near the north of the domain and over the east-draining watersheds.

Contrary to Figure 11, heterogeneity significantly influences local subsurface flow by affecting how water moves through and interacts with different soil layers, as evidenced in Figure 12. Variations in soil properties, such as texture and permeability, impact water retention and infiltration rates. Land use and vegetation cover alter infiltration and evapotranspiration processes,

while atmospheric conditions like precipitation and temperature affect the overall water budget in the soil column and runoff generation. Figure 13 differentiates the subsurface flow between stream and land pixels. Heterogeneous land cover changes the mean by ~56%, reducing the water that reaches the stream and river network. In comparison, the contributing land also experiences an overall reduction of outflow of ~70%, which could be influenced by higher water uptake and evaporation due to vegetation cover. Similar behaviour is shown under the heterogeneous soil properties and meteorology experiments.

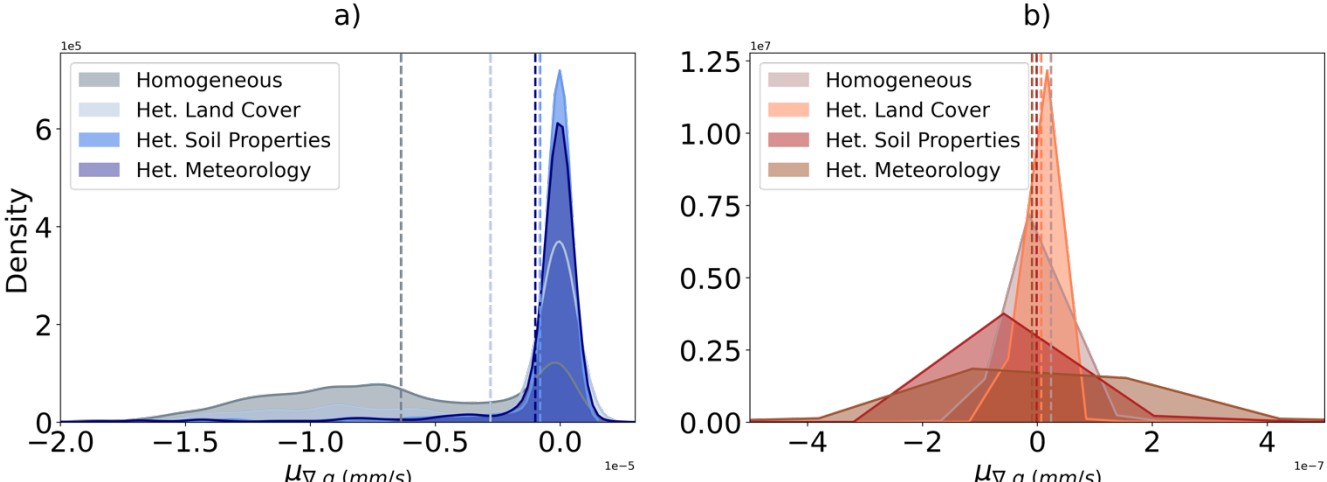


**Figure 13: Distribution of divergent subsurface flow for streams (a) and land pixels (b) showing the change in spatial mean (dotted lines) for the different heterogeneous scenarios.**

The heterogeneous case presented in Figure 12c was used to highlight the differences between the proposed multiscale scheme for simulated subsurface lateral flow and the current implementation on HydroBlocks, which only simulates the local

subsurface flow.

As shown in Figure 14, four output variables of the LSM were chosen to compute the differences against the baseline (only local lateral flow): soil moisture content (SMC), latent heat flux (LH), sensible heat flux (SH), and surface runoff (R). The maps presented here represent the year-long hourly mean for the last year of a 50-year hourly simulation and are discussed below.



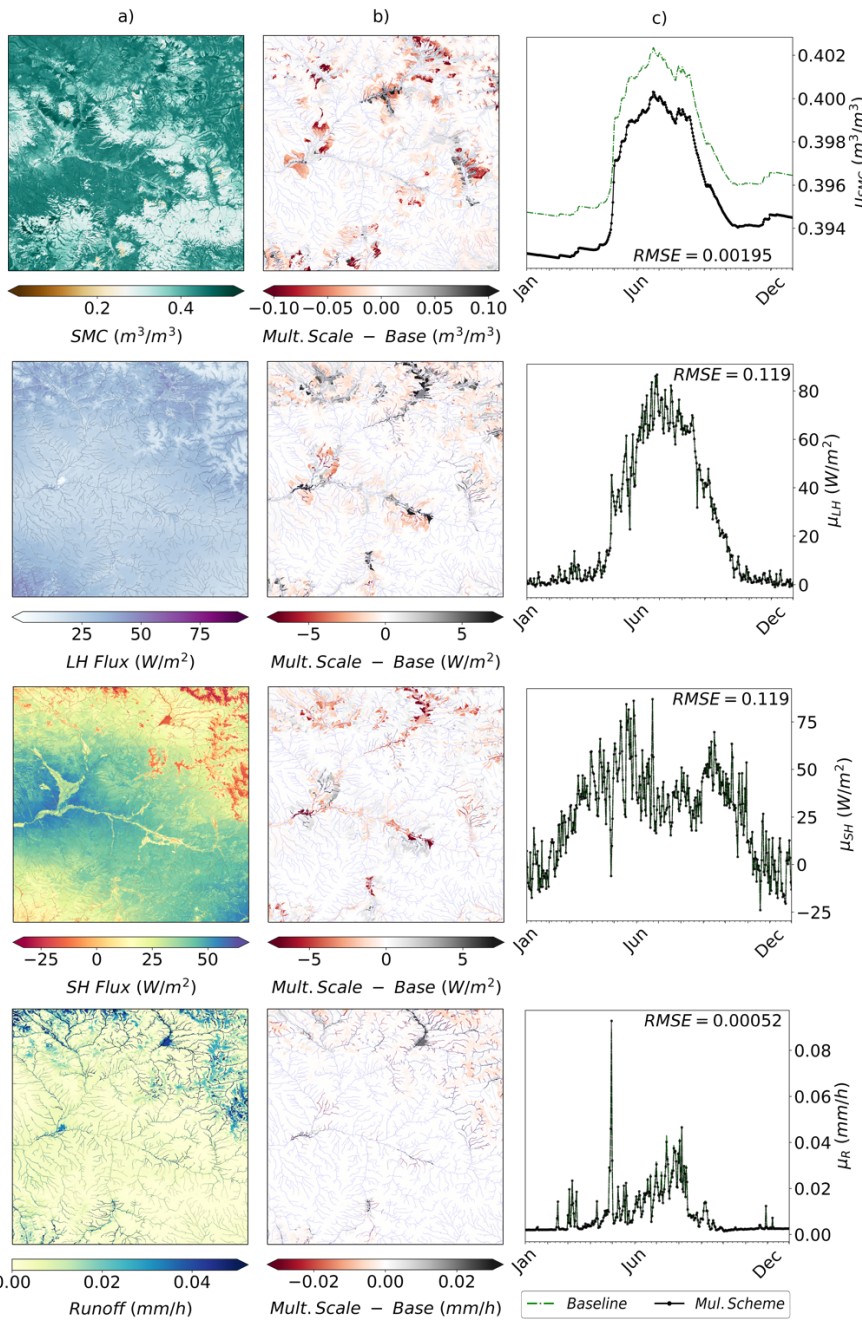

**Figure 14: Comparison between the enhanced multiscale scheme in HydroBlocks and the baseline for SMC, LH, SH, and R. Column (a) shows the spatial distribution of the time-averaged values for each variable. Column (b) displays the difference between the multiscale scheme and the baseline. Column (c) illustrates the daily mean time series for the last year of a 50-year simulation for each variable over the entire study domain.**

*Soil Moisture Content (SMC):* Figure 14 presents the map of SMC for the 10 m deep soil column (column a). Compared to the baseline, the map (column b) reveals positive differences predominantly in the valley areas, while negative





differences appear on the ridges. In some places, the RISFU used in the simulation can be identified as areas of uniform water gain or water loss. The time series of the spatial average for the last year of simulation shows an overall decrease in magnitude while still capturing the temporal variability.

The observed positive differences in valley areas can be attributed to enhanced water accumulation in these lower-lying regions due to a redistribution of flows throughout our regional scheme. In contrast, the negative differences on the ridges are due to faster drainage and reduced water retention, as these elevated areas are more prone to water loss. The uniform patterns of water gain and loss in the RISFU areas highlight the influence of regional to intermediate hydrological processes and soil characteristics for those particular spatial structures.

*Latent and Sensible Heat Flux (LH/SH):* The Latent and Sensible Heat Flux maps presented in Figure 14 (column a) follow a similar pattern to that of SMC. In areas with relatively high subsurface flow, more water is available near the surface for evaporation, increasing the latent heat flux. Conversely, as more water moves through the soil, carrying heat with it, either cooling or warming the soil and altering the temperature gradient between the surface and the air, sensible heat flux decreases or increases respectively.

Valley areas in the domain experience a positive difference in latent heat flux (and a comparable negative difference in sensible heat flux) due to the relatively higher soil moisture levels than the baseline. Subsurface flows can be more limited on ridges due to steeper topographic gradients. Consequently, the impact on latent and sensible heat fluxes is less pronounced in these areas, where higher temperatures and lower soil moisture levels prevail. However, recharge areas can be identified near streams, often occurring at relatively higher elevations than the locations they ultimately drain into.

The temporal variability of the spatial average for both energy fluxes closely matches that of the baseline (refer to column c) in Figure 14. This strong correlation is primarily attributed to the dominant processes occurring in the superficial soil layers, which directly affect evaporation and heat exchange with the atmosphere. These layers respond rapidly to changes in atmospheric conditions and exert a significant influence on both components of the energy budget. In contrast, while subsurface flow redistributes soil moisture and heat within deeper layers, its impact on surface energy fluxes is less pronounced. Changes

in moisture and heat within deeper layers take longer to propagate to the surface, and their effects are attenuated by the overlying soil layers.

*Surface Runoff (R):* Compared to the baseline, the new multiscale scheme shows a positive difference in surface runoff for streams in lower elevation areas. This suggests more consistently saturated conditions resulting from contributing subsurface flow from riparian zones and higher elevation watersheds, as shown in columns (a) and (b) of Figure 14. A distinct

pattern is observed in the northeastern corner of the domain, where smaller watersheds exhibit lower runoff rates in the drainage areas and higher flow rates in the streams immediately downstream. In contrast, for the major rivers in the central part of the domain, the impact on runoff generation is relatively dampened due to the influence of dominant regulatory processes. In this region, meteorological factors play a more significant role in runoff generation, as evidenced in Figure 11c. The time series presented in column (c) of Figure 14 illustrates the temporal variation of spatially averaged runoff, indicating no major changes

in the seasonality.



### 3.3 Approximation to the benchmark simulation

The outputs of the long-term simulation from the previous experiment, which involved heterogeneous conditions, were compared against a HydroBlocks benchmark. This benchmark consisted of 1.4 million tiles distributed into 64 parallel processes. We compared the long-term hourly mean for the last year of our 50-year simulation of the four output variables:

SMC, LH, SH, and R. Figure 15 shows both the multiscale scheme in column (a), the benchmark simulation in column (b) and a difference map (a) - (b) in column (c). For each variable, the mapped temporal mean shows a reasonable agreement between the results of the multiscale scheme and the benchmark simulation regarding variable range and spatial pattern distribution, as evidenced by the similarity in spatial mean and spatial standard deviation.

Nonetheless, the differences presented in column (c) of Figure 15 appear to indicate a more even distribution of variables near

the center of the domain. The multiscale scheme seems to better capture the location and dynamics of streams and river networks, which is very evident for the LH, SH, and R fields. There are a few potential reasons why the multiscale scheme appears to capture river locations better, while the benchmark simulation shows a more uniform distribution in those areas.

The aggregation process in the multiscale scheme –as a result of a lower number of tiles– could introduce biases that emphasize larger hydrological features by reducing noise in areas with smaller features. The benchmark simulation, being more sensitive

to local variations, might not exhibit this bias. For example, in the lower-resolution model, each tile aggregates the hydrological response over a larger area, effectively smoothing spatial variability and emphasizing dominant features such as rivers. This averaging effect can make streams more apparent because the model blends high and low values over a larger spatial extent, highlighting the influence of persistent hydrological features.

In contrast, the higher-resolution simulation resolves finer-scale variability, potentially distributing water across multiple

smaller tiles rather than concentrating it in river channels. This can lead to an artificial spreading of water across neighboring areas, reducing localized concentration along the river network.





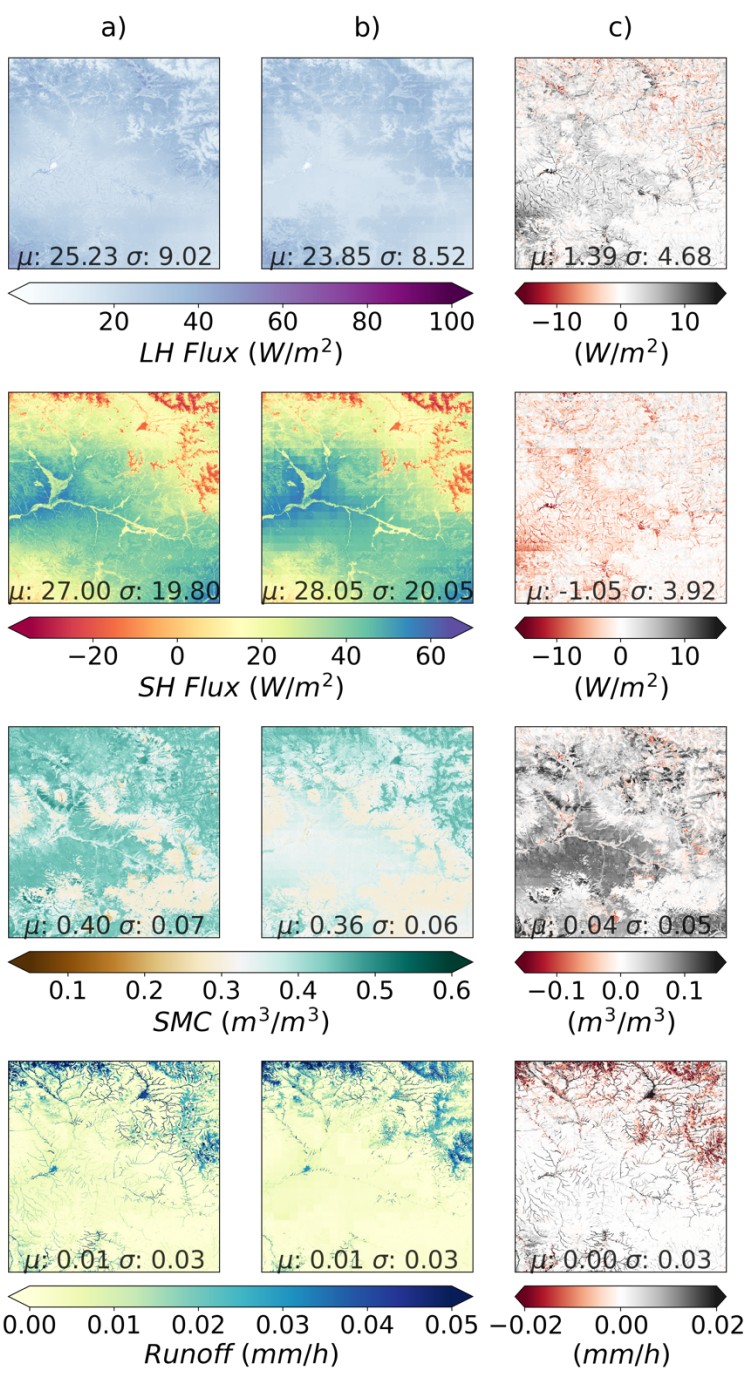

**Figure 15: Output comparison of HydroBlocks Multiscale Scheme with 40.700 tiles (column a) and benchmark simulation with 1.4 million tiles (column b). The spatial fields of SMC, LH, SH, and R are hourly averages of the one-year-long output time series. Column (c) shows the difference (a) - (b). The grid-like pattern in maps of column (b) is attributed to the imprint of the meteorological forcing dataset at 1/32° resolution compared to domain subdivision at 1/64°**






To further assess the performance of the multiscale scheme relative to the benchmark simulation, Figure 16 presents a comparison of the daily averages of the spatial mean and standard deviation for hydrological variables, SH, LH, R, and effective saturation ($\Theta$). The results indicate a strong overall agreement in the temporal variability of SH and LH between both

models, as demonstrated by high Pearson correlation coefficients for both the mean and standard deviation time series. However, there are evident discrepancies during the summer months, a period when subsurface flow becomes a dominant hydrological process, particularly in low-elevation river areas.

These seasonal differences are further accentuated in the R and $\Theta$ time series. The multiscale scheme differs by around ~20% from the runoff generated in the benchmark simulation, with higher rates in the multiscale scheme. In the case of effective

saturation, the temporal mean values closely align between the two simulations, indicating consistency in large-scale water balance representation. However, the standard deviation time series indicates a larger variability of saturated conditions throughout the domain for the benchmark.

This increased variability further confirms the discrepancies observed in the spatial distribution maps shown in Figure 15. The multiscale scheme provides a more detailed representation of streams and river networks than the benchmark simulation.

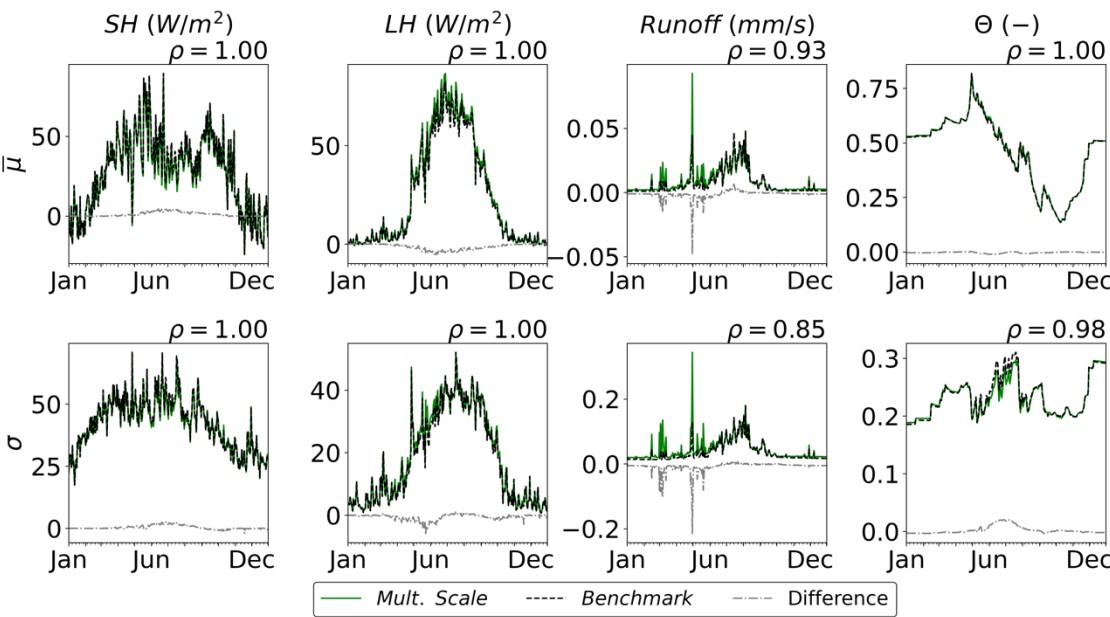


**Figure 16: Temporal variability of the spatial mean ($\mu$) and standard deviation ($\sigma$) fields for LH, SH, R, and $\Theta$. In green (—) is the multiscale scheme variable output; dash black line (- - -) represents the results of the benchmark simulation. The grey line (—·—) is the difference between the multiscale scheme and the benchmark model. $\rho$ is Pearson's correlation coefficient.**

At this stage of analysis, a persistent difference between the two simulations can be observed, which is likely attributed to the

representation of small-scale topographic heterogeneity in the benchmark simulation and a scale mismatch in the multiscale scheme. These differences indicate that the way elevation variability is represented might not be equally captured by the current



tile-based configuration in both models. To further investigate these differences, we analyzed the temporal variability of SMC at two different depths of the soil column.

Figure 17 presents a comparative analysis of SMC at the root zone level (left column) and the bottom of the soil column (right column). At the root zone level, the agreement between the multiscale scheme and the benchmark simulation is strong, as indicated by a high Pearson correlation coefficient for both the mean and spatial standard deviation values. This close alignment suggests that the model effectively captures the dynamic processes occurring within the root zone, where soil moisture fluctuations respond rapidly to external influences such as root water uptake, evaporation, and precipitation events.

However, at deeper soil levels, substantial differences are introduced in both the mean and spatial standard deviation values. These discrepancies highlight the challenges associated with modeling deeper soil layers. In these deeper layers, the effects of elevation become increasingly significant, influencing water distribution and movement. Unlike the root zone, where near-surface processes primarily drive water flow, deeper layers are subject to more diffuse and less direct flow paths, with pressure gradients influenced by subtle variations in elevation that may not be equally represented in both models and that larger tiles accentuate elevation gradients leading to faster subsurface flows reaching streams areas earlier compared to the benchmark simulation.

In contrast, the benchmark simulation, with its finer representation, may predict a distribution of states, resulting in a more homogenized spatial distribution of subsurface flows instead of a single representative state for the equivalent of a single tile. These findings emphasize the importance of improving the representation of elevation heterogeneity in the tiling approach for both simulations to ensure a more robust validation of the multiscale scheme.

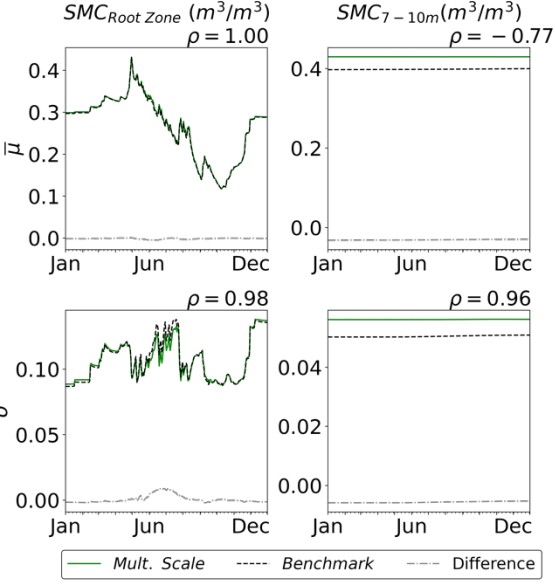

**Figure 17: Temporal variability of the spatial mean ($\mu$) and standard deviation ($\sigma$) fields for SMC at root zone (left column) and bottom of the soil column (right column). In green (—) is the multiscale scheme variable output; dash black line (---) represents the results of the benchmark simulation. The grey line (–·–) is the difference between the multiscale scheme and the benchmark model. $\rho$ is Pearson's correlation coefficient.**



## 4 Discussion

### 4.1 Advanced Subsurface Flow Simulation in LSMs Using the HydroBlocks Model

This study introduced a new scheme for simulating subsurface lateral flow in LSMs at local, intermediate, and regional scales, leveraging a novel and efficient tiling framework within the HydroBlocks LSM model. This approach aims to address the underrepresentation of subsurface flow paths between neighboring watersheds that could influence small-scale hydrological processes within large-scale domains while leveraging high-spatial and temporal resolution data and preserving computational efficiency.

A major advantage of the current scheme is that it is designed for implementation in parallel computing systems. The lateral interaction among regional and intermediate subsurface flow units introduced here can be easily implemented to exploit the benefits of parallel simulations, and it can be scaled up to accomplish continental scale simulations following the same principles for inter-domain communication. An idea that is not new and that has already been implemented in the routing scheme in HydroBlocks (Chaney et al., 2021), allowing for a more integrated hydrological system. Despite not being used in this study, HydroBlocks is capable of simulating channel flow across macroscale polygons and inter-domain communication. This capability highlights future directions for evaluating the impact of regional and intermediate subsurface flow on surface water distribution at the field scale, addressing a common problem in representing local processes influenced by regional groundwater dynamics.

The results of this study show a clear difference between the enhanced HydroBlocks simulation using the proposed multiscale scheme and the baseline case. These differences showcase the advantages of our approach to represent the multiple scales required for a more comprehensive simulation of the subsurface dynamics and potentially the groundwater system.

### 4.2 Parameterization Challenges and Validation in Multiscale Hydrological Modeling

Though not the focus of this study, it is important to acknowledge the uncertainties associated with the datasets used, which certainly can translate to biases in model predictions when compared to actual observations. Additionally, as the resolution of the model increases, an inherent uncertainty is introduced to the model processes due to the extensive requirements for model parameterization (Beven et al., 2015; Bierkens et al., 2015). We compared spatial and temporal variability of generated time series of soil moisture content, latent and sensible heat flux, as well as runoff generation and effective saturation with a more discretized solution (benchmark) of our test domain using HydroBlocks, for a 50-year hourly simulation. Our validation shows agreement between the multiscale scheme and the benchmark in terms of spatial mean, standard deviation, and Pearson's correlation coefficient; this highlights the capacity of our scheme to capture seasonal variability and average spatial dynamics while still preserving computational efficiency. When considering the parameterization of the LSM, the proposed approach relies on the redistribution of soil moisture in the vertical soil column and the lateral flow between them driven by a modeled hydraulic gradient; such characterization of the soil column and land surface properties obey a single realization of the datasets presented in Section 2.1. However, a sensitivity analysis of the parameters could lead to a more comprehensive understanding



of its role in the subsurface dynamics to approach a closer representation of observed fields. This is important as models such as HydroBlocks, and consequently, the introduced multiscale scheme permit thousands of simulations to explore parameter sensitivity and uncertainty quantification (Chaney et al., 2016).

One area of improvement for the current model structure is the representation of soil column depth. Our current assumption of a constant soil column depth of 10 meters everywhere neglects groundwater flows occurring at greater depths. Importantly, this undermines the role of subsurface heterogeneity, as one would expect shallower vadose zones near the lowland and deeper in the upland. To address these issues, future work should focus on: i) altering the parameterization to include *variable soil column depths and rock structure* involving detailed characterization of vadose zone across different topographical features

and ii) *characterizing aquifer properties at high spatial resolutions*, incorporating variations in hydraulic conductivity, porosity, and storage coefficients. This could be accomplished by *integrating diverse datasets*, including geological surveys, remote sensing data, and detailed soil surveys, to refine the model inputs and reduce uncertainties.

When validating high-spatial-resolution models at the macroscale, point-based observations often struggle to adequately capture the spatiotemporal dynamics of hydrological variables (Koch et al., 2015). Simultaneously identifying both spatial and

temporal patterns is crucial, especially as the current generation of LSMs and computational resources advance toward more detailed representations of land surface dynamics (Koch et al., 2016; Torres-Rojas et al., 2024; Zink et al., 2018). These patterns are essential for validating distributed models and predictions by effectively capturing trends and anomalies in environmental data. For example, flooding dynamics are highly sensitive to local factors such as channel bathymetry, riverbed slope, and floodplain inundation, making extensive observations critical for validating water level dynamics. The launch of

the Surface Water and Ocean Topography (SWOT) mission provides high-resolution (~100 m) water surface elevation data, offering a unique opportunity to study flooding dynamics and enhance their representation in LSMs (Biancamaria et al., 2016; Pedinotti et al., 2014). By validating HydroBlocks against SWOT data, we could improve our understanding of the processes governing flooding dynamics and refine both LSM structures and spatially distributed validation strategies.

**4.3 Balancing Efficiency and Accuracy in HydroBlocks Pseudo-3D Approach for Lateral Flow**

The proposed methodology is flexible, allowing its implementation in large-scale simulations while still preserving computational efficiency by leveraging the tiling-based structure characteristic of HydroBlocks. However, some assumptions and limitations are inherent to this approach.

In HydroBlocks, a pseudo-3D numerical solution of Richards' equation decouples vertical and lateral water flow simulation to reduce computational complexity while preserving important dynamics (Chaney et al., 2021). Vertical flow is solved first in

Noah-MP, capturing processes like infiltration and root water uptake under unsaturated/saturated conditions. Then, lateral flow is computed based on the hydraulic gradient between adjacent soil columns of the modeling tiles. This iterative approach allows for the interaction of vertical and lateral components without requiring a fully simultaneous 3D solution, making it computationally efficient while still representing both vertical and lateral groundwater flows in the model.



However, this structure relies on several assumptions: First, vertical and lateral flows occur on different time scales, with vertical movement being much faster than lateral. This usually holds where steep topographic gradients and unsaturated conditions allow gravity to predominate as the acting force. The second assumption is a quasi-steady state lateral flow that, along with the hydrostatic assumption, simplifies the numerical solution of Richards' equation for flat terrains and antecedent dry soil conditions, where the lateral flow becomes less significant. Finally, the assumption of small horizontal gradients permits lateral flow to be treated as a secondary process.

The assumption of small horizontal gradients might not hold at deeper soil layers. Complex interactions and variations in pressure and hydraulic gradients due to elevation differences become more significant, making it harder for the model to capture lateral flow accurately. This could explain why the model performs better at the root zone than deeper soil layers.

In natural settings like agricultural fields or shallow aquifers, vertical processes such as infiltration, evaporation, and root uptake are typically faster and more critical to resolve. At the same time, lateral flow often occurs more slowly and can be approximated using hydraulic gradients. This makes the pseudo-3D solution a practical alternative to fully 3D models, which, although more accurate, are computationally expensive.

### 4.4 Optimal Spatial Representation of Subsurface Structure in Hydrological Simulations

This study employs clusters of watersheds to define regional and intermediate modeling units, and during the experimental phase, watersheds were also tested, yielding similar results. However, when watersheds are used as modeling units, the spatial disconnection between tiles becomes problematic for accurately redistributing regional flow back to individual tiles, as one tile can belong to two different watersheds. Despite this limitation, the numerical solution and computational efficiency gained from this approach make it worthwhile. The model balances accuracy and processing speed by focusing on a simplified regional structure, enabling simulations at high spatial resolutions that would otherwise be computationally limited.

To better represent the subsurface structure, it is important to account for aquifer heterogeneity, which can significantly affect water flow and storage. Incorporating detailed geological and hydrogeological data into the model can enhance accuracy rather than relying solely on clusters of watersheds or individual watersheds as defining parameters for the subsurface structure. This involves integrating information about varying soil types, rock formations, and aquifer properties to more precisely capture the spatial variability in hydraulic conductivity, porosity, and other key parameters for groundwater flow.

Incorporating these detailed subsurface characteristics allows for a more realistic simulation of groundwater flow and interactions between different layers of soil and rock. This enhanced representation helps address the limitations of assuming a constant distance to bedrock or treating regional units as a replica of superficial water divisions.

As shown in Section 3.3: Approximation to the benchmark simulation, the influence of topography over lateral water redistribution between tiles is highly dependent on the tile configuration. Addressing the challenges associated with optimal spatial representation of land surface heterogeneity is required. Moreover, the optimal tile configuration is model-dependent, and one could prioritize subsurface heterogeneity over surface heterogeneity in natural settings where groundwater dynamics is the major driving mechanism; this is clustering based on soil properties and elevation.



## 5 Conclusions

This study evaluates the potential of using a tiling scheme to represent regional and intermediate flow paths using HydroBlocks LSM. The LSM enables the high-resolution simulation of water and energy fluxes within the landscape. The proposed

methodology comprises the definition of parameterized regional and intermediate subsurface flow units using the soil hydraulic properties of the tiles within each unit to define the dynamic movement of water between soil columns. Soil hydraulic properties such as soil moisture content, saturated and residual water content, and hydraulic conductivity are crucial for accurately simulating water flow. The experiment adopts a constant soil column depth and resolves the vertical water movement using Noah-MP as part of the HydroBlocks framework.

The current approach adopts a clusters of watersheds configuration for the regional and intermediate modeling units to represent the subsurface structure. It defines the lateral interaction between them across subdomains to exploit the advantages of parallel computing systems. This configuration mimics natural hydrological boundaries, ensuring water flow simulation across different regions.

The current study demonstrates the potential of the tiling scheme within a defined domain. It also lays the path to implementing

this multiscale scheme at continental scales, which requires addressing boundary conditions across domains to reflect the continuity of subsurface flow paths and the influence of large natural hydraulic drivers, such as lakes and oceans, on regional groundwater dynamics.

Due to computational limitations, the methodology is compared against a quasi-fully distributed solution of HydroBlocks of 1.4 million tiles. This benchmark simulation allows for detailed spatial representation while balancing computational

efficiency, making it a suitable benchmark for comparison. The multiscale scheme converges to the benchmark in spatial mean and standard deviation, as well as temporal variability, indicating the potential, robustness, and accuracy for simulating regional hydrological processes. However, more work is required to validate the proposed methodology, as the elevation heterogeneity representation in both models differ considerably, and the benchmark simulation appears to homogenize the lateral water redistribution, while the multiscale scheme simulates faster subsurface flow that reaches streams and river

networks faster compared to the benchmark.

*Code and data availability:* Data files and code for model simulations are available at Guyumus, D., & Chaney, N. (2025). Supporting Dataset HydroBlocks-MSSUBv0.1: A Multiscale Approach for Simulating Lateral Subsurface Flow Dynamics in Land Surface Models (Version 1) [Data set]. Zenodo. https://doi.org/10.5281/zenodo.14825442

*Author contributions:* NWC, LTR, and CX developed the model. DG developed and implemented the new scheme, performed

the simulations and analysis, and prepared the first draft of the manuscript. All authors contributed to the design of the study, revision, and writing of the final version.

*Competing interest:* At least one of the (co-)authors is a member of the editorial board of Geoscientific Model Development.

*Acknowledgments:* This study was supported by funding from NOAA grants NA23OAR40504311I, "Next-Generation of NOAA water modeling: Climate Risks & Interactive Sub-seasonal to Seasonal Predictability in the Earth System modeling



framework—Bipartisan Infrastructure Law/Infrastructure Investment and Jobs Act," and NA24OARX431C0052-T1-01, "Confronting the GFDL land model's sub-grid tiling scheme with observed space-time patterns of land surface temperature: Implications for hydrologic extremes."

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
