# Peer review of "HydroBlocks-MSSUBv0.1: A Multiscale Approach for Simulating Lateral Subsurface Flow Dynamics in Land Surface Models"

_EGUsphere, 2025_

## Referee Comment (RC1)

**Review for "HydroBlocks-MSSUBv0.1: A Multiscale Approach for Simulating Lateral Subsurface Flow Dynamics in Land Surface Models" by Daniel Guyumus et al.**

First, I would like to congratulate the authors for this paper and the original approach it is based on. They propose their solution to a hotly debated topic in the community : How are we going to represent lateral transports through the various aquifers in our land surface models while having to cope with relatively coarse resolution atmospheric forcing.

Obviously no study or paper is perfect and I have a few questions and suggestions. I encourage the authors to think about them when preparing a revised version.

Before coming to scales of decomposition of hydrological structures, there is one point which needs to be considered. This is the physical assumptions on the separation of vertical and horizontal water movements. This is not well described or justified in the manuscript. A graphic illustrating how over the 10m of simulated soils the vertical and horizontal fluxes are positioned would be very helpful. What is particularly worrying to me is that in the unsaturated zone the Richards equation is applied while we do not know where the Darcy law is used. Based on their respective assumptions, the Darcy law should only be applied to the water in the saturated portion of the 10m soil column. How this is organized in the model should be clearly explained to the community as it is at the core of the question of how to decompose the 3D movements of water. Do we need to go to a full 3-dimensional resolution of the Richards equation or can a clever decomposition of the horizontal and vertical dimensions work ? I would have a problem if the Darcy law is also applied to the unsaturated part of the soil !

The decomposition of the hydrological structures according to the scales is original and very effective. The entire paper shows how this facilitates the representation of the lateral flows. But to my knowledge there is no clear scale separation of the surface hydrological structures. So what is the basis for separating the regional, intermediate and tiled components ? From the graphics it is obvious that the regional units are atmospheric grids. What would happen if instead of being at about 25km resolution the atmospheric forcing would be at resolutions of 2.5 or 250km (some typical resolutions of atmospheric models used today) ? Is the decomposition numerically robust to other shapes for the polygons of the atmospheric grid (triangles or hexagons) ?

Generally I find the approach to represent the fluxes between RISFUs quite original and convincing. But I fail to understand how the hydraulic head differences are computed. Is it just the elevation difference between RIFUs or is the water table depth established in each soil column to compute the hydraulic gradients ? If it is the second option, then I would like to know which criteria is used to define the water table depth in each soil column. It would reassure me if Darcy is only used for the flux in the saturated zone and not in the unsaturated part.

Minor comments :
- line 90 : 1820 core-hours is not a pertinent metric as it depends on the type of cores used and the software environment. Relative differences are more meaningful.
- Figure 1 : Would it not be helpful to put references to the datasets used for the illustrations ?
- Line 124 : The observational product for soil types are described relative to the 10m soil depth of the model. More relevant would be to know which information on the vertical distribution of soil properties is available before projected onto the vertical discretization. Later then how this information is interpreted/used by the soil moisture model.
- Line 124 : How are the 10m of soil depth chosen ? Is this the observed bed rock depth over the region or is it arbitrary ?
- Line 141 : I have the impression that it is the clustering method which does the scale separation for you. It would be interesting to determine how the clustering methods respond to the choice of macroscale polygons. Here probably lies the answer to my second general concern expressed above.
- Line 183 : Please remind the reader here about the difference in hypothesis between the Richards and Darcy equations. The Darcy law does not take into account the forces which bound the liquid to the porous material and thus can only be applied in a saturated medium.
- Line 283 : It should only be done for the layers which are saturated.
- Line 341 : Is there not a case as well where the atmospheric conditions are assumed homogeneous ?

- Line 393 : It is not clear to me if in your scheme the rivers can also lose water to the surrounding riparian zones. This is a particularly important process when rivers cross shallow aquifers.
- Figure 13 : Write that you are using μ to designate means and σ variances. Your notation is not so common and thus can be confusing. Using them as functions could also help.
- Figure 14 : The green lines are not visible in all subplots of column c. Either explain in the legend that they are hidden beneath the black line or find another representation.
- Figure 14 : Why runoff and not discharge ? You are not illustrating the delays induced by the flow through the shallow aquifers which should have an important impact.
- Line 509 : Why introduce here effective saturation when volumetric soil moisture is the prognostic variable ? The variable is not clearly defined either. Is this the same thing as soil wetness, which more common in LSMs ?
- Line 536 : I do not understand the argument why the coarser model will accentuate gradients and then produce different fluxes. The distances are larger as well. The coarser model will have more numerical issues and thus generate different fluxes. But at this stage we do not know where the numerical challenges are at representing the subsurface flows. More generally we do not know which numerical scheme is used to solve the Darcy equation.
- Line 536 : Here the argument is made that the river discharge should be examined. Why is this not used as a metric earlier in this study ?
- Line 585 : I suspect this discussion will need to be revised as I believe the depth of the water table will be much more critical. Above the Richards equation can be applied with all its numerical difficulties. Below the Darcy flows can be used an simplifying the numerical solution. But as the authors note, above the water table vertical movement are dominant and thus there is no need to solve a 3D Richards equation.
- Line 614 : In the saturated zone the lateral flows are faster than the vertical ! In the vadoze zone it is the contrary.

---

## Referee Comment (RC2)

Reviewer's comments for Guyumus et al. 2025: "HydroBlocks-MSSUBv0.1: A Multiscale Approach for Simulating Lateral Subsurface Flow Dynamics in Land Surface Models"

General Summary:
In this study, the authors present a novel approach to compute lateral subsurface flow across multi-scale (local, intermediate, and regional) using HydroBlocks coupled in NoahMP LSM. The simulations are setup in a 1deg x 1deg box in western Colorado for a 50-year long-term simulation to test the model efficiency in capturing spatial heterogeneity and temporal variability of the multiscale approach, compared to a high-resolution tiling (1.4 million tiles) for benchmarking. The results show consistent spatial distribution and high temporal variability of the multiscale approach with the high-resolution benchmark in capturing latent heat, sensible heat, soil moisture and runoff.

This novel approach tackles a challenging research problem to accurately represent subgrid heterogeneity and lateral subsurface flow, while also advance in computational efficiency, providing potential revenue towards large-scale/continental simulation. I recognize the substantial amount of work dedicated to this manuscript and accomplishment, but I do have a few comments to make, please see my comments below:

1. Although this study addresses the lateral subsurface flow for groundwater, the computation of local, intermediate and regional flow in Step4 and Step5 seems to add up to soil moisture for that particular layer and column. In particular in Step5, it seems the HydroBlock is only updating soil moisture content to NoahMP - does HydroBlock also update water table depth to NoahMP? That said, there isn't an explicit definition of water table depth (WTD) for each column, separating unsaturated and saturated soil moisture, from this scheme. This is important because the Richard equation would apply to the unsaturated zone while Darcy equation to the saturated zone.

2. In Step2, L223 "the number of RISFUs is determined by the parameter k chosen equal to 20". How is this k parameter determined for this domain? Does it depend on elevation variation? In this case, the domain elevation (Figure1b) exhibits large variation - what about running a simulation over the Great Plains region or along the Mississippi river, where elevation variation could be small and most resolving subgrid microtopography such as potholes, and the water table depth could be very shallow (close to surface)?

3. The performance of HydroBlock multiscale approach - how much does it depend on the resolution of the NoahMP column model? I could have missed it, what's the resolution of the NoahMP column model, is it the same as the meteorological forcing at 1/32 deg (roughly 3km). How is the model performance different from the original NoahMP column model at its own resolution? That said, is it possible to compare the simulation results (multiscale approach, benchmark high-resolution tiling, and original NoahMP grid), it would be curious to see if the multiscale approach improves the spatial distribution compared to the original NoahMP grid?

4. Last but not least, this study focuses on the impacts of resolving subgrid heterogeneity of soil moisture. The authors could also discuss the effect of better resolving this heterogeneity to influence land-atmosphere interactions in regions where groundwater is

shallow, topography heterogeneity is critical (river valley), see Barlage et al. (2021) "The Importance of Scale-Dependent Groundwater Processes in Land-Atmosphere Interactions Over the Central United States" (https://doi.org/10.1029/2020GL092171) and Zhang et al. (2025) "US Corn Belt enhances regional precipitation recycling" (https://doi.org/10.1073/pnas.2402656121) that connecting subsurface groundwater heterogeneity to soil moisture and atmospheric water cycle, precipitation recycling, etc., especially in convection-permitting atmospheric models.

Specific comments:
L134: This should be 2.2
Figure14c: It seems the multiscale and baseline are identical for LH, SH and R, while their difference is large for soil moisture. Are these because the range of the variable, the range for SMC is smaller while LH and SH range are large? Also want to make sure that HydroBlock is updating SMC to NoahMP and then LH/SH/R are computed from NoahMP (with updated SMC)?
L365: 3Results, it would be good for the authors to provide a table to summarize the three simulations conducted in this study - 3.1 homogeneous baseline run, 3.2 heterogeneous conditions run, and a high-resolution tiling benchmark run. I was a bit confused between the baseline and benchmark run. Also it would be good to present NoahMP results without HydroBlock (maybe in supplementary materials).
L530: Please add a clear definition of the root-zone soil moisture, is it 0-1m deep?

---

## Author Comment (AC2)

We thank the reviewers for their feedback and contributions to the improvement of this manuscript. We have revised the manuscript with a new set of simulations, following the correction of a bug in our code that was reducing the soil hydraulic conductivity. We have now updated the figures and the corresponding analysis.

Please find the response to your comments in blue.

Reviewer #1

Review for "HydroBlocks-MSSUBv0.1: A Multiscale Approach for Simulating Lateral Subsurface Flow Dynamics in Land Surface Models" by Daniel Guyumus et al.

First, I would like to congratulate the authors for this paper and the original approach it is based on. They propose their solution to a hotly debated topic in the community: How are we going to represent lateral transports through the various aquifers in our land surface models while having to cope with relatively coarse resolution atmospheric forcing.

Obviously no study or paper is perfect and I have a few questions and suggestions. I encourage the authors to think about them when preparing a revised version.

Before coming to scales of decomposition of hydrological structures, there is one point which needs to be considered. This is the physical assumptions on the separation of vertical and horizontal water movements. This is not well described or justified in the manuscript. A graphic illustrating how over the 10m of simulated soils the vertical and horizontal fluxes are positioned would be very helpful. What is particularly worrying to me is that in the unsaturated zone the Richards equation is applied while we do not know where the Darcy law is used. Based on their respective assumptions, the Darcy law should only be applied to the water in the saturated portion of the 10m soil column. How this is organized in the model should be clearly explained to the community as it is at the core of the question of how to decompose the 3D movements of water. Do we need to go to a full 3-dimensional resolution of the Richards equation or can a clever decomposition of the horizontal and vertical dimensions work? I would have a problem if the Darcy law is also applied to the unsaturated part of the soil!

We thank the reviewer for this relevant comment. We agree that a clear explanation of how vertical and horizontal fluxes are treated in the model is required to understand the physical assumptions of the proposed decomposition approach. We have updated Section 2.3 and include Section 2.4 to discuss this.

In our model, vertical water movement within each soil column is governed by the one-dimensional Richards equation, accounting for the hydraulic conductivity as a nonlinear function of the water content in the soil layer.

For lateral water movement between neighboring soil columns, we apply Darcy's law directly, even in unsaturated conditions. However, we ensure that the hydraulic conductivity used in this formulation is fully dependent on the local soil moisture status. Specifically, lateral flow is calculated using pressure head gradients between corresponding soil layers of adjacent columns, and the hydraulic conductivity is computed using the same unsaturated retention and conductivity functions (Brooks-Corey) that are used in the vertical direction. This ensures that lateral fluxes diminish appropriately in dry soils.

We understand the importance of this modeling assumption, specifically application of Darcy's law in saturated porous media traditionaly assumes no vertical flow in most LSM. Our model does not requires this to be the case, and it can adjust a separation between saturated and unsaturated zones to ensure lateral flow only happens in saturated conditions. However, the simulations presented here do no account for such distinction.

We also added a brief discussion of the limitations of this approach, including scenarios where it may introduce error due to strong lateral gradients in Section 4.3

The decomposition of the hydrological structures according to the scales is original and very effective. The entire paper shows how this facilitates the representation of the lateral flows. But to my knowledge there is no clear scale separation of the surface hydrological structures. So what is the basis for separating the regional, intermediate and tiled components? From the graphics it is obvious that the regional units are atmospheric grids. What would happen if instead of being at about 25km resolution the atmospheric forcing would be at resolutions of 2.5 or 250km (some typical resolutions of atmospheric models used today)? Is the decomposition numerically robust to other shapes for the polygons of the atmospheric grid (triangles or hexagons)?

We have included Section 4.5 to discuss this.

Regarding the physical and numerical justification for the chosen scales and geometries, we agree that in nature, the boundaries between hydrological processes operating at different scales are not clearly defined. We have introduced the scale separation to manage complexity and reduce the computational demands while isolating dominant subsurface processes.

This decomposition is not based on a fixed natural boundary but rather an approximation that aligns hydrological processes with their characteristic spatial scales. In our case, the scale decomposition is as follows:

- Tile/local scale: used to resolve topographic, land cover, and soil properties heterogeneity.
- Intermediate scale: Represents dominant lateral flow pathways between watersheds, allowing for lateral redistribution within physical bounds, within the same macroscale polygon.
- Regional scale: Chosen to match lateral interaction between physical units that resemble atmospheric grid cells while allowing lateral interaction between parallel simulations.

The current decomposition indeed uses a subdivision of the macroscale polygons, specifically the cluster of watersheds, as regional units. However, we emphasize that the approach is resolution-independent in principle.

If atmospheric forcing is provided at finer resolution (here we are using 3 km resolution PCF), the regional units can be adjusted accordingly, and the intermediate and tile structures would adjust hierarchically within these smaller domains.

For coarser resolutions, intermediate units may aggregate differently, possibly requiring more internal sub-resolution structure to maintain hydrological realism. We would explore this sensitivity in future work to investigate the accuracy of lateral flux representation and the dependence of surface responses on how well intermediate and regional processes are resolved.

Regarding the robustness of the approach, the numerical framework is designed to be flexible with respect to the shape of the regional units, as long as a consistent topological and geometric representation is provided. In our implementation, lateral flow computations and flux redistributions are formulated in a generalized way that can accommodate different polygonal grids. Nonetheless, the shape of the RISFU will influence the realism of specific lateral pathways, especially near boundaries or in regions with strong directional flow patterns.

Generally, I find the approach to represent the fluxes between RISFUs quite original and convincing. But I fail to understand how the hydraulic head differences are computed. Is it just the elevation difference between RIFUs, or is the water table depth established in each soil column to compute the hydraulic gradients? If it is the second option, then I would like to know which criteria are used to define the water table depth in each soil column. It would reassure me if Darcy is only used for the flux in the saturated zone and not in the unsaturated part.

We have updated Section 2.5. Step 4 to clarify the computation of Hydraulic Heads. We also included Figure 6 to add clarity.

In our model, lateral fluxes between RISFUs are computed using a Darcy-type flow where the hydraulic gradient is defined as the difference in total hydraulic head between adjacent RISFUs, divided by the horizontal distance between them. The hydraulic head is defined as the sum of the pressure head and the elevation of the contributing layer. To define the pressure head for lateral flux, we use the local soil moisture conditions at a specific layer in each RISFU and the elevation of that layer.

We do not explicitly compute the water table position to define hydraulic gradients, the saturation-dependent transmissivity limits lateral flow in unsaturated conditions.

Minor comments:

• Line 90: 1820 core-hours is not a pertinent metric as it depends on the type of cores used and the software environment. Relative differences are more meaningful.

We have adjusted the paragraph in the same line.

• Figure 1: Would it not be helpful to put references to the datasets used for the illustrations?

We added the references in the caption. Lines 120-134 include the description to the respective datasets.

• Line 124: The observational product for soil types are described relative to the 10m soil depth of the model. More relevant would be to know which information on the vertical distribution of soil properties is available before projected onto the vertical discretization. Later then how this information is interpreted/used by the soil moisture model.

We have clarified the use of POLARIS in Section 2.1

• Line 124: How are the 10 m of soil depth chosen? Is this the observed bed rock depth over the region or is it arbitrary?

We have included the following clarification:

"Although the chosen depth of the soil column is mostly arbitrary, we consider 10 meters sufficient to evaluate the influence of lateral subsurface flows over land surface processes. 10 m is not a strict or universal cutoff, but we have adopted it as a reasonable approximation according to previous studies (Fan et al., 2013; Kollet and Maxwell, 2008; Shokri-Kuehni et al., 2020; Whittington and Price, 2006), where it has been shown that land-surface processes are considered not water-limited when the water table is near the surface, as capillary rise can supply water to the root zone, reducing water stress for plants. And conversely, surface processes are viewed as water-limited when the water table is deep, as they rely entirely on precipitation."

• Line 141: I have the impression that it is the clustering method which does the scale separation for you. It would be interesting to determine how the clustering methods respond to the choice of macroscale polygons. Here probably lies the answer to my second general concern expressed above.

Yes, the scale separation in HydroBlocks is performed through the combination of macroscale polygons and hierarchical clustering within them. The macroscale polygons are user-defined spatial domains that provide a structure for parallel computation and a geographic container for applying the HMC algorithm to derive the tiles of hydrologic similarity.

The choice of macroscale polygons can influence the clustering outcome, as they determine the upper boundary of the subsequent clustering decomposition. However, since this is not part of the scope of the present work, we believe that we can investigate this further in future work.

• Line 183: Please remind the reader here about the difference in hypothesis between the Richards and Darcy equations. The Darcy law does not take into account the forces which bound the liquid to the porous material and thus can only be applied in a saturated medium.

We have included a clarification in line 243

• Line 283: It should only be done for the layers which are saturated.

We have included a description of our assumptions in section 2.4. We are not restricting lateral flow to only saturated layers.

• Line 341: Is there not a case as well where the atmospheric conditions are assumed homogeneous?

We have improved the representation of this scenario in Figure 13. We do account for homogeneous meteorological forcing in the Homogenous experiment, Heterogenous Land cover and Heterogeneous Soil Properties.

• Line 393: It is not clear to me if in your scheme the rivers can also lose water to the surrounding riparian zones. This is a particularly important process when rivers cross shallow aquifers.

The experiment presented here does not account for river routing. A new version of HydroBlocks includes a more efficient scheme for river routing across macroscale polygons. Future work includes validating this scheme with river discharge over a large domain.

• Figure 13: Write that you are using μ to designate means and σ variances. Your notation is not so common and thus can be confusing. Using them as functions could also help.

Thanks for pointing this out. We have updated figures and included in captios and text the meaning of the symbols.

• Figure 14: The green lines are not visible in all subplots of column c. Either explain in the legend that they are hidden beneath the black line or find another representation.

We have updated our figures improve clarity

• Figure 14: Why runoff and not discharge ? You are not illustrating the delays induced by the flow through the shallow aquifers which should have an important impact.

We are considering runoff generation as our experiment does not include river routing.

• Line 509: Why introduce here effective saturation when volumetric soil moisture is the prognostic variable? The variable is not clearly defined either. Is this the same thing as soil wetness, which more common in LSMs?

This is very valid. We have modify the figures to only discuss the now 5 output variables and replace effective saturatio for soil moisture content to avoid confussion.

• Line 536: I do not understand the argument why the coarser model will accentuate gradients and then produce different fluxes. The distances are larger as well. The coarser model will have more numerical issues and thus generate different fluxes. But at this stage we do not know where the numerical challenges are at representing the subsurface flows. More generally we do not know which numerical scheme is used to solve the Darcy equation.

We have updated Section 3.3. Our argument is that the benchmark simulation better resolves small scale gradientes while the multiscale simulation can damp small scale heterogeneity with a single realization of subsurface fluxes while benchmark will represent the entire distribution. The new simulations appear to have resolved this problem.

• Line 536: Here the argument is made that the river discharge should be examined. Why is this not used as a metric earlier in this study?

We do not include river routing in this experiment.

• Line 585: I suspect this discussion will need to be revised as I believe the depth of the water table will be much more critical. Above the Richards equation can be applied with all its numerical difficulties. Below the Darcy flows can be used an simplifying the numerical solution. But as the authors note, above the water table vertical movement are dominant and thus there is no need to solve a 3D Richards equation.

We have include clarification on the groundwater scheme used in NoahMP for computing water table. We have also included analysis of water table elevation and updated the results accordingly. We have included Section 2.4 to present the assumptions made.

• Line 614: In the saturated zone the lateral flows are faster than the vertical! In the vadoze zone it is the contrary.

We have added the clarification.

Reviewer's comments for Guyumus et al. 2025: "HydroBlocks-MSSUBv0.1: A Multiscale Approach for Simulating Lateral Subsurface Flow Dynamics in Land Surface Models"

General Summary:

In this study, the authors present a novel approach to compute lateral subsurface flow across multi-scale (local, intermediate, and regional) using HydroBlocks coupled in NoahMP LSM. The simulations are setup in a 1deg x 1deg box in western Colorado for a 50-year long-term simulation to test the model efficiency in capturing spatial heterogeneity and temporal variability of the multiscale approach, compared to a high-resolution tiling (1.4 million tiles) for benchmarking. The results show consistent spatial distribution and high temporal variability of the multiscale approach with the high-resolution benchmark in capturing latent heat, sensible heat, soil moisture and runoff.

This novel approach tackles a challenging research problem to accurately represent subgrid heterogeneity and lateral subsurface flow, while also advance in computational efficiency, providing potential revenue towards large-scale/continental simulation. I recognize the substantial amount of work dedicated to this manuscript and accomplishment, but I do have a few comments to make, please see my comments below:

1. Although this study addresses the lateral subsurface flow for groundwater, the computation of local, intermediate and regional flow in Step4 and Step5 seems to add up to soil moisture for that particular layer and column. In particular in Step5, it seems the HydroBlock is only updating soil moisture content to NoahMP - does HydroBlock also update water table depth to NoahMP? That said, there isn't an explicit definition of water table depth (WTD) for each column, separating unsaturated and saturated soil moisture, from this scheme. This is important because the Richard equation would apply to the unsaturated zone while Darcy equation to the saturated zone.

In our implementation, lateral subsurface fluxes across the three scales (local, intermediate, and regional) are computed in HydroBlocks and returned to Noah-MP as source/sink terms to the soil moisture content of each vertical column.

Regarding water table depth, we do not explicitly define or update a WTD in the standard sense, nor do we impose a hard separation between saturated and unsaturated layers. Instead, we rely on the 1D Richards equation used in Noah-MP to simulate vertical water movement under variable saturated conditions. We have added Section 2.4 to clarify our assumptions.

WTD dynamics depend on Noah-MP's parameterization, which relies on the simulated vertical soil moisture profile. Our multiscale lateral flow scheme does not override this; rather, by modifying the soil moisture profile through lateral fluxes, we enable the model to adjust the WTD. We have updated Section 2.3 to describe the groundwater scheme adopted within NoahMP.

For lateral flow, we apply a Darcy-type formulation even under unsaturated conditions, but we use soil moisture-dependent hydraulic conductivity derived from Brooks-Corey functions. This ensures that in dry soils, the lateral conductivity and lateral flow are minimal, while maintaining a continuous representation of subsurface redistribution without explicitly segmenting saturated vs unsaturated layers.

We acknowledge that this is a key assumption and have revised the manuscript to clarify. Our model can accommodate the separation between saturated and unsaturated if needed. We discussed this in section 4.3

2. In Step2, L223 "the number of RISFUs is determined by the parameter k chosen equal to 20". How is this k parameter determined for this domain? Does it depend on elevation variation? In this case, the domain elevation (Figure1b) exhibits large variation – what about running a simulation over the Great Plains region or along the Mississippi river, where elevation variation could be small and most resolving subgrid microtopography such as potholes, and the water table depth could be very shallow (close to surface)?

We added clarification Line 325: "In our study, the value of k=20 was chosen empirically to capture the spatial heterogeneity of watershed-scale features while maintaining computational efficiency over a domain with significant elevation variability. The clustering step in HydroBlocks' HMC uses K-means over proxies for large-scale physical heterogeneity (latitude, longitude, elevation, flow accumulation area). Thus, the parameter k does not reflect variability on elevation alone but the interaction of multiple spatial drivers."

In flatter regions, such as the Great Plains or the Mississippi River basin, where elevation gradients are small, fine-scale soil differences or the presence of potholes, the choice of k may need to be adjusted. A smaller k could suffice in capturing large-scale homogeneity, while in some cases, a large k or the inclusion of additional covariates may better represent the spatial heterogeneity. Future work involves in validating this scheme, including river routing over different watersheds in the US.

3. The performance of HydroBlock multiscale approach - how much does it depend on the resolution of the NoahMP column model? I could have missed it, what's the resolution of the NoahMP column model, is it the same as the meteorological forcing at 1/32 deg (roughly 3km). How is the model performance different from the original NoahMP column model at its own resolution? That said, is it possible to compare the simulation results (multiscale approach, benchmark high-resolution tiling, and original NoahMP grid), it would be curious to see if the multiscale approach improves the spatial distribution compared to the original NoahMP grid?

To clarify, HydroBlocks does not replace the land surface modeling component; instead, it defines the hydrologically similar units (tiles) over which land surface processes are resolved. In this study, each tile is simulated using the Noah-MP column model, which computes the energy and water balances. We have modified Section 2.3 to clarify this.

The Noah-MP model works at the scale of tiles defined by HydroBlocks, which are generated from high-resolution (~30 m) data and grouped into representative units using the Hierarchical Multivariate Clustering (HMC) scheme. Meteorological forcing data is available at approximately 3 km resolution, with each tile using the average values from the meteorological grid cells it covers.

We agree that a quantitative comparison between the original Noah-MP grid-based setup and the HydroBlocks tiling approach would be highly valuable. However, such a comparison is beyond the scope of the current study, but we have simulated the same domain without lateral interactions. This means, in principle, that the model is resolved using native Noah-MP with a tile discretization. From this set of simulations, we observed that our multiscale scheme accumulates more water in the lowlands and channels compared to the simulation with no lateral connections.

[Figure]

[Figure]

4. Last but not least, this study focuses on the impacts of resolving subgrid heterogeneity of soil moisture. The authors could also discuss the effect of better resolving this heterogeneity to influence land-atmosphere interactions in regions where groundwater is shallow, topography heterogeneity is critical (river valley), see Barlage et al. (2021) "The Importance of Scale-Dependent Groundwater Processes in Land-Atmosphere Interactions Over the Central United States" (https://doi.org/10.1029/2020GL092171) and Zhang et al. (2025) "US Corn Belt enhances regional precipitation recycling" (https://doi.org/10.1073/pnas.2402656121) that connecting subsurface groundwater heterogeneity to soil moisture and atmospheric water cycle, precipitation recycling, etc., especially in convection-permitting atmospheric models.

We appreciate the reviewer's suggestion. We have now included a reference to this point in the revised manuscript (introduction), noting the significance of lateral subsurface flow at high spatial resolution.

"In alignment with the findings of Barlage et al. (2021), finer spatial resolution enables better representation of topographic convergence zones and shallow water table dynamics. Their study demonstrates that resolving these features at convection-permitting scales leads to increased root-zone soil moisture, enhanced evapotranspiration, and improved feedbacks to the atmosphere, including reductions in temperature and precipitation biases."

Specific comments:

L134: This should be 2.2

Thanks for pointing this out. We have adjusted this line.

Figure14c: It seems the multiscale and baseline are identical for LH, SH and R, while their difference is large for soil moisture. Are these because the range of the variable, the range for SMC is smaller while LH and SH range are large?

We identified a bug in our code that was damping the hydraulic conductivities of the layers. We have now updated the figures, yielding similar results. However, there is a clear difference in SMC, R, and water table elevation (ZWT) due to a redistribution of subsurface flows. LH and SH are still almost identical, but the range in differences can be appreciated in column b of Figure 15. This is in fact relatively small compared to the range of SMC.

Also want to make sure that HydroBlock is updating SMC to NoahMP and then LH/SH/R are computed from NoahMP (with updated SMC)?

Yes. HydroBlocks is updating soil moisture through the sink/source term in the differential equation resolved by NoahMP.

L365: 3 Results, it would be good for the authors to provide a table to summarize the three simulations conducted in this study - 3.1 homogeneous baseline run, 3.2 heterogeneous conditions run, and a high-resolution tiling benchmark run. I was a bit confused between the baseline and benchmark run. Also it would be good to present NoahMP results without HydroBlock (maybe in supplementary materials).

Thanks for this suggestion. We have included a summary at the end of Section 3. Results.

However, we have not included a only-Noah-MP simulation as we consider this outside of the scope of the present work. HydroBlocks has been developed to leverage high-spatial resolution data to better resolve the role of small-scale heterogeneity. While a fully distributed model is possible, the computational cost of resolving <100 m resolution grid cells is impractical for large-scale simulations. Chaney et al., (2016) demonstrated the convergence of the clustering scheme to the fully distributed solution with a fraction of the computational cost.

L530: Please add a clear definition of the root-zone soil moisture, is it 0-1m deep?

We have added the clarification. It is a 0-5 cm.

Reviewer #3

General comments:

This work is highly valuable for the land surface modeling community because 1) improving the representation of subsurface hydrology remains a critical challenge for LSMs, and 2) computational efficiency must be carefully considered especially when applying to larger scale applications. This multiscale approach addresses both needs by enhancing subsurface hydrology representation while maintaining computational feasibility, which is commendable. The manuscript is well written with sufficient details in methodology and experimental results, but I do have a few comments, which I hope the authors could address to provide a clearer picture of how the proposed approach may benefit the community.

Specific comments:

1. L117: Is a 40-year spin-up sufficient for the groundwater system to reach equilibrium? Please include a justification for this chosen spin-up duration.

Thanks for pointing this out. We consider 40 years of spin-up sufficient for this experiment, based on the convergence of the mean SMC, SH, and ZWT; however, this may be site-specific. Here we present the comparison of spatial mean for the multiscale simulation and benchmark simulation, showing the change after each iteration. The percentage of change between the last two iterations is less than 2%.

Multiscale Scheme 40700 tiles

[Figure]

Benchmark simulation 1.4 million tiles

2. L166-168: Since Noah-MP can simulates recharge, water table change and baseflow vertically, it is unclear why these processes are only highlighted when combined with HydroBlocks. Please clarify.

Thanks for pointing this out. We do not mean to suggest that these processes are 'only' highlighted when used with HydroBlocks. We have revised the sentence to clarify that HydroBlocks employs Noah-MP to calculate the energy and water balances for each tile. This means HydroBlocks determines the modeling units over which Noah-MP will perform water and energy balance computations. We have updated section 2.3 to clarify this.

3. L222-225: Is the choice of the parameter k sensitive to the size of the study domain and/or the size of the macroscale polygon? For global-scale application, what considerations should be made

in selecting k and polygon sizes? Additionally, how would these choices affect computational cost?

Yes. The parameter k selection affects how surface heterogeneity is depicted and the proxies used for its measurement. There is no established method to determine the optimal number of tiles needed to accurately represent heterogeneity in hydrologic processes within a macroscale grid cell or across various spatial scales, but Torres-Rojas et al. (2022) have addressed this issue previously. They introduced a metric to identify the best parameter configuration that captures heterogeneity more effectively while minimizing computational costs of a fully distributed simulation. However, this remains an area for further research. HydroBlocks has not yet been used globally; current work includes simulations over CONUS and Finland, though further validation is necessary.

Regarding computational cost for the Multiscale Scheme: For reference, a parameter k = 60 was selected for the benchmark simulation, resulting in approximately 1.4 million tiles, increasing the computational costs by a factor of 8.5. We, however, need to indicate that the total number of tiles depends on the combination of parameters k, n, and p, and not only k.

Additionally, regardless of the value of parameter k, the multiscale scheme presented here can accommodate diffent configurations for the RISFU units, thus they do not require to be the same as the cluster of watersheds.

We added clarification in Line 325: "In our study, the value of k=20 was chosen empirically to capture the spatial heterogeneity of watershed-scale features while maintaining computational efficiency over a domain with significant elevation variability. The clustering step in HydroBlocks' HMC uses K-means over proxies for large-scale physical heterogeneity (latitude, longitude, elevation, flow accumulation area). Thus, the parameter k does not reflect variability on elevation alone but the interaction of multiple spatial drivers."

4. L342-343: Why do the authors not include the validation for groundwater levels? Given that the lateral subsurface flow scheme likely influences groundwater recharge and can be remarkable for areas with high elevation difference and with intensive human water management, this variable is critical – particularly for applications in regions with intensive groundwater pumping for irrigation.

We appreciate this important comment and fully agree that groundwater dynamics is important to assess the multiscale scheme, particularly in regions with significant topographic gradients and intensive groundwater extraction. However, the goal of the current study is to demonstrate the feasibility of a new lateral subsurface flow scheme within LSMs, with an emphasis on representing interactions across spatial scales. At this proof-of-concept stage, our validation focuses on internal model consistency and convergence in a controlled experimental setup, rather than on direct comparison with observations.

That said, we recognize the importance of validating groundwater levels, particularly for applications in hydrologically active or human-managed basins. In future work, we aim to evaluate the model over a larger, more heterogeneous domain to better assess regional subsurface processes and their hydrological connectivity. For such an evaluation, single-point groundwater observations may not provide sufficient spatial context. Instead, we intend to rely on spatially distributed metrics and remote sensing datasets (e.g., GRACE, SWOT) that may be better suited for capturing integrated hydrological responses over larger extents.

We also note that the current implementation does not yet include river routing. This is a planned extension, as it will be necessary for linking groundwater and surface water dynamics and for validating against streamflow time series at the basin scale.

5. I think it would be beneficial to include a comparison to Noah-MP without the tiling framework. This would provide a clearer picture of how the proposed modeling framework compared to the native Noah-MP simulation without lateral flow, in addition to its comparison with the finer HydroBlocks benchmark. Such comparison would inform regional or global modelers considering lateral flow integration.

Thanks for this suggestion. While comparing HydroBlocks against native Noah-MP is very relevant, the computational cost of simulating an equivalent high-resolution model (~30m) over this domain is beyond the purpose of this analysis. Since HydroBlocks does not replace the land surface modeling component, but instead, it defines the modeling units (tiles) over which the land surface processes are resolved. We have ran a simulation without lateral connection, that in principle is the native NoahMP over a discretized domain. From this set of simulations, we observed that our multiscale scheme accumulates more water in the lowlands and channels compared to the simulation with no lateral connections.

[Figure]

[Figure]

6.  Figure 12c: Could the authors clarify why the spatial pattern of subsurface flow divergence does not follow the west-east precipitation gradient as described in Figure 1? Although Figure 1 shows more like a climatology representation while this comparison only focuses on a single year, it might be helpful to display the precipitation distribution for the specific validation year alongside this subplot to illustrate how meteorological forcing affects subsurface flow divergence.

We have updated Figure 1 to reflect the spatial pattern in precipitation. We have also updated Figure 13 (previous Figure 12) to better represent the spatial pattern in subsurface flow. From this, the spatial patterns of both precipitation and divergent subsurface flow are closely related.

7.  Figure 14: It is unclear why the impact of spatially averaged seasonality on soil moisture in the multiscale approach appears much greater than that on runoff. I would expect there is large difference on runoff as well, especially under the assumption of baseline-controlled meteorology. Can the authors elaborate on this result?

Thanks for this comment. We have identified a bug in our code that was significantly reducing hydraulic conductivity of the layers, leading to less generated runoff. We have updated our simulations and identified that now runoff, soil moisture and water table elevation are the variables with major differences with respect to the baseline. We have updated Figure 15.

8. L518-519: Are there in-situ observations available that could be used to evaluate whether the multiscale approach yields improved representation of streams and river networks compared to the benchmark?

We thank the reviewer for this question. At this stage of model development, we have not included a river routing scheme, as our goal is to evaluate the internal behavior and feasibility of the proposed multiscale scheme in isolation. Including river routing would introduce additional sources of complexity and feedback. As this is a proof of concept, we aimed to isolate and better understand the effects of lateral flow parameterized across spatial scales. For the same reason, our validation strategy is designed to test internal model consistency rather than performance against external observations. Specifically, we validated our multiscale implementation against a quasi-fully distributed version of the same domain. This comparison allows us to evaluate whether the proposed approach can replicate hydrological patterns without the computational burden of resolving fine-scale lateral connectivity everywhere.

We recognize the value of evaluating stream and river networks, particularly in larger-scale applications. In future work, once river routing is incorporated and the model is applied to a larger domain, we plan to compare simulated discharge and streamflow dynamics with available in-situ observations and remote sensing products. These comparisons will be essential for assessing how well our approach can reproduce the connectivity and organization of subsurface dynamics.

9. Since one of the goals for the development of this approach is to preserve computational efficiency, it would be helpful to provide detailed information on the computational cost of the multiscale approach relative to the HydroBlock benchmark, as well as the Noah-MP simulation. Perhaps section 4.3 could be expanded to discuss performance metrics. Moreover, offering suggestions/comments on how this multiscale approach could be applied in regional (e.g., CONUS) or global simulations – including appropriate resolutions and parameter recommendations – would enhance its practical relevance. Currently, the discussion lacks clarity on how the multiscale approach compares computationally with either HydroBlocks or traditional 1-D LSMs like Noah-MP.

We have updated Section 4.3 to explicitly include a comparison of the computational cost between the multiscale simulation and the benchmark. As indicated in the revised text, the benchmark simulation requires approximately 8.5 times more core hours than the multiscale configuration over the same domain, demonstrating a substantial gain in computational efficiency.

We agree that assessing the computational scalability and optimal setup of the multiscale approach over large regions, such as CONUS or even global areas, is essential for broader application. However, a detailed investigation of regional scale implementation is beyond the scope of this proof-of-concept study. Ongoing work focuses on applying and fine-tuning the multiscale scheme over larger domains, including a successful implementation over the Connecticut River Basin, which has enabled us to start examining scaling behavior and how it interacts with the river routing scheme.

While our current study does not include specific recommendations for resolutions or parameterizations at regional or global scales, we envision that the multiscale approach could serve as a middle ground between computationally expensive fully distributed models and overly simplistic 1D LSM. Future work will aim to assess how best to configure the scheme for broader domains and the inclusion of river routing.

10. L566-568: It remains unclear to me whether the multiscale approach shows advantage in simulating subsurface dynamics and groundwater system as, 1) there is no evaluation against observations and 2) there is no investigation on the simulation for groundwater storage or water table depth. The authors may consider providing additional insights or evidence to earn this point.

We agree that assessing the multiscale approach's capability to represent subsurface and groundwater dynamics is essential for evaluating its utility.

Although this study primarily aims to demonstrate a proof-of-concept for lateral subsurface interactions across scales, we have taken additional steps to clarify this aspect.

First, we have expanded the manuscript to include an analysis of the water table elevation variable, which is directly influenced by the multiscale lateral flow scheme. In particular, we discuss how water table depth is represented and parameterized within Noah-MP and how this interacts with our new lateral flow formulation. These additions are now included in Section 2.3.

Second, we acknowledge that the current study does not include evaluation against in-situ groundwater observations or GRACE-derived groundwater storage estimates. This decision was made intentionally at this stage, as our simulations do not yet include surface water routing, and the goal is to isolate the topography-driven lateral subsurface signal. However, we agree this is a critical next step, especially in regions where subsurface-surface interactions are important.

We have begun applying the multiscale scheme to larger domains such as the Connecticut River Basin, which will enable more robust evaluation using both groundwater level observations and spatially distributed remote sensing datasets. These future applications will allow us to assess the integrated performance of the multiscale model in representing groundwater storage, streamflow, and water table variability in more complex hydrological settings.

---

## Author Response (AR2)

We thank the Editor and reviewers for their feedback. We believe increasing the spinup time has significantly improved the simulations. We have addressed the comments in the following manner:

"I concur with one reviewer's concern regarding spin-up equilibrium: the multiscale scheme exhibits larger interannual variability in SMC and ZWT compared to the benchmark even after 40 years. Please provide time series showing evolution of domain-averaged SMC and ZWT over the full spin-up period for both simulations, and clarify whether this variability reflects genuine process differences or insufficient equilibration."

We have extended our simulations to a total of 150 years and included Figure 1 in the manuscript. We found this very valuable, and the reviewers were right to bring this up. We have addressed this comment and believe it enhances our simulation significantly. We also ran a simulation using the same 10 years of forcing data. The results show a state of equilibrium for the benchmark simulation, while the Multiscale simulation appears to reach a dynamic equilibrium. This is evident in Figure 1, where water table elevation (ZWT) fluctuates around the mean despite the convergence in soil moisture content (SMC) and latent heat flux (SH). We have included this figure as Figure 10 in the main manuscript. The oscillations observed in the Multiscale scheme, as well as in the baseline simulation, may indicate issues related to scale representation in both domain discretizations, which could lead to greater variability in the spatial mean of water table elevation, or numerical instability within the groundwater scheme in NoahMP, influenced by our model's lateral flows. Further analysis is needed to explore the compatibility between the numerical solution and the scale representation in HydroBlocks; however, this is beyond the scope of this study (see Section 4.3, line 748).

[Figure]

Figure 1. Comparison of estimated soil moisture content, sensible heat flux, and water table elevation throughout the spin-up period for Baseline (40,700 tiles), Multiscale Scheme (40,700 tiles), and Benchmark (1.4 million tiles).

"Another reviewer requests clearer documentation of applying Darcy's law with moisture-dependent conductivity across saturated and unsaturated zones—please expand Section 4.3 to discuss conditions under which the hydrostatic equilibrium assumption breaks down and clarify whether this approach is fundamental to the implementation or a simplifying convenience."

We have included a new section, "4.4 Suitability of Unsaturated Lateral Flow in the Multiscale Scheme," to clarify that our implementation is a simplification for representing unsaturated lateral flow at hyper-resolutions. However, it can accommodate the saturated-only lateral flow if necessary.